# Impact of Oil Temperature and Splashing Frequency on Chili Oil Flavor: Volatilomics and Lipidomics

**DOI:** 10.3390/foods14061006

**Published:** 2025-03-16

**Authors:** Xiaoping Li, Xiaopeng Liu, Shiting Su, Zhao Yao, Zhenhua Zhu, Xingyou Chen, Fei Lao, Xiang Li

**Affiliations:** 1College of Culinary and Food Science Engineering, Sichuan Tourism University, Chengdu 610100, China; 2College of Food Science and Nutritional Engineering, National Engineering Research Center for Fruit & Vegetable Processing, Key Laboratory of Fruit and Vegetable Processing of Ministry of Agriculture and Rural Affairs, China Agricultural University, Beijing 100083, China; 3School of Health Industry, Sichuan Tourism University, Chengdu 610100, China; 4College of Food and Biological Engineering, Qiqihar University, Qiqihar 161006, China; 5School of Food Science and Technology, Dalian Polytechnic University, Dalian 116034, China

**Keywords:** red oil, aroma, triglycerides, fatty acids

## Abstract

In this study, headspace gas chromatography–ion mobility spectrometry, headspace gas chromatography–mass spectrometry, and lipidomics were used to explore the effects of three oil temperatures (210 °C, 180 °C, 150 °C) with single- and traditional triple-oil-splashing processes (210 °C → 180 °C → 150 °C) on the formation of key chili oil aromas. A total of 31 key aroma compounds were identified, with 2,4-nonadienal, α-pinene, α-phellandrene, and β-ocimene being found in all treatment groups. Lipidomics suggested that oleic acid, linoleic acid, and α-linolenic acid were highly positively correlated with key chili oil key aroma compounds, such as (E)-2-heptenal, 2-methylbutyraldehyde, limonene, (E, E)-2,4-heptadienal, 2,4-nonadienal, and 2,4-decadienal. The temperature and frequency of oil splashing significantly affected the chili oil aroma profile (*p* < 0.05). The citrus, woody, and grassy notes were richer in chili oil prepared at 150 °C, malty and fatty aromas were more prominent at 180 °C, and the nutty aroma was stronger in 210 °C prepared and triple-splashed chili oil. The present study reveals how sequential oil splashing processes synergistically activate distinct lipid degradation pathways compared to single-temperature treatments, providing new insights into lipid-rich condiment preparation, enabling chefs and food manufacturers to target specific aroma profiles.

## 1. Introduction

Many cuisines in China use chilis and their products [1,2,3]. Sichuan cuisine is one example, known for its development of the distinctive red oil flavor [4], which is based on an authentic chili oil preparation. Red oil, also known as Sichuan chili oil, is a condiment prepared by heating vegetable oil to a specific temperature and pouring it over chili powder [5]. It is used to add a signature flavor to Sichuan dishes [6]. Therefore, the flavor of chili oil is an important factor affecting its application and acceptance.

Previous studies on the effects of oil temperature and chili varieties on the aroma compounds of chili oil found that aldehydes, alcohols, ketones, and esters, represented by 2-methylbutanal, octanal, (E)-2-octenal, nonanal, hexanal, 3-methylbutanal, 1-octen-3-ol, dihydro-2 (3H) -furanone, 1-hydroxy-2-acetone, and ethyl acetate were the main volatile flavor substances in chili oil, which provided fatty, citrus, nutty, floral, and caramel aromas to the chili oil [5,7]. The flavor of chili oil is significantly correlated with the pepper variety and vegetable oil type; for example, chili oil made from Mogui and Zidantou peppers is hexanal-rich with apple and grassy aromas, that made from Shizhuhong and Erjingtiao peppers has more nonanal with a fatty and citrus aroma, and that made from Xiaomila pepper contains more (E, E)-2,4-nonadienal with fried and fatty aromas [5]. Chili oil made from soybean oil has a strong citrus and fruity aroma, whereas that made from peanut oil has a rich fatty and nutty aroma [8]. Chili oil made from rapeseed oil tends to have more fatty, floral, and spicy aromas [8]. The particle size of chili powder significantly affects the quality of chili oil, with reduced particle size (e.g., 60 mesh) enhancing the dissolution rate of capsaicinoids and optimizing product quality [9,10]. Among environmental factors, high temperature, light exposure, and oxygen accelerate lipid oxidation, particularly leading to the degradation of terpenoids and the production of aldehydes and ketones volatiles. Sealing, light avoidance, and low-temperature storage are necessary to delay flavor deterioration [11,12,13]. Furthermore, storage time is negatively correlated with the retention rate of key flavor compounds (such as limonene and linalool), indicating the need for controlled shelf life to maintain flavor stability [12].

The oil splashing temperature and process also affect the flavor of the chili oil. The contents of aldehydes and terpenes in chili oil are significantly affected by the production of chili oil at 120, 140, 160, 180, and 200 °C [14]. Researchers have also set different oil splashing temperature gradients and let them sit for 30 min; their results suggested that at 160 °C, the oil contained more aldehydes, acids, and olefins than at 140 °C and 150 °C, and that the aroma quality was best at this temperature [15]. However, these investigations largely overlooked the dynamic thermal choreography inherent to authentic Sichuan practices—specifically, the triphasic oil splashing at descending temperatures (210 °C → 180 °C → 150 °C) followed by 24 h infusion. Thermal oil extraction induces lipid oxidation, characterized by hydroperoxide formation and degradation into flavor-active compounds such as aldehydes, ketones, alcohols, acids, and furans [16]. Research indicates that immersing chili oil in beef tallow for 24 h can enhance the capsaicin content by 15.8% [17], while immersing five-spice flavored oil for over 18 h significantly promotes the dissolution of volatile flavor compounds [18]. Although prolonging the immersion time can enhance the flavor quality, extending it beyond 24 h results in an increased peroxide value [19]. Given that under normal temperature, lipids remain in the initial stage of oxidation (with a lower peroxide value) within 24 h [20], precisely controlling the immersion time of chili oil to 24 h can both maximize the dissolution of flavor compounds and effectively inhibit excessive oxidation. Therefore, collecting the aroma profiles after 24 h of extraction would provide a more realistic flavor experience for chili oil. In addition, comparing the variation in volatiles among single-oil-splashing temperatures as well as classical Sichuan triple oil splashing, together with 24 h oil extraction, would provide valuable information to decode the preparation of authentic Sichuan red oil, thereby providing a better understanding of how the signature Sichuan flavor is formed. Although previous studies have conducted preliminary research on the factors affecting the flavor of chili oil, how traditional multi-stage heat treatment coordinates the complex chemical processes behind the unrepeatable aroma of Sichuan chili oil is still unknown.

Chili powder contains proteins, fats, and reducing sugars [20], which are prone to lipid oxidation and Maillard reactions under heating conditions, forming various volatile organic compounds [21]. During this process, proteins undergo thermal degradation and participate in Maillard reactions, generating flavor compounds such as pyrazines and pyrroles. The increase in nitrogen-containing compounds (such as pyrroles) found in the study by Ye [22] could possibly originate from the thermal degradation of proteins. Meanwhile, heating also leads to extensive degradation of fatty acids [23]. During the frying process, unsaturated fatty acids are broken down into aldehydes, ketones, furans, and pyridine compounds [20], and they interact with Maillard reaction products [24]. In addition, carbohydrates (such as reducing sugars) undergo caramelization at high temperatures (140–180 °C), producing furans, aldehydes (such as isovaleraldehyde and 2-methylbutanal), and other key aroma compounds in chili oil [15]. The formation of typical flavor substances under oil thermal conditions is generally related to the Maillard reaction, Maillard–lipid interactions, and lipid metabolism [25,26]. As a branch of metabolomics, lipidomics is a powerful tool for elucidating lipid composition, analyzing key lipid markers, and identifying critical metabolic pathways [27,28]. Lipidomics adds great value to food quality and flavor understanding owing to its efficient and comprehensive ability to detect shifts in lipid composition in complex food substrates. In recent years, lipidomics has been actively used in flavor metabolism to reveal the lipid-involved characteristic flavor formation pathways. It has been applied in the analysis of lipid fingerprints of air-dried salmon at different temperatures [29], the characterization of lipid and flavor changes in salted goose jerky fermentation [30], and understanding the evolution of the non-volatile flavor of fresh tea [31]. We pioneer its application in the analysis of Sichuan chili oil flavors, and the involvement of lipidomics would add new chemical insights into Sichuan chili oil decoding.

The objective of this study was to investigate the effect of different oil splashing temperatures (210 °C, 180 °C, and 150 °C) and oil splashing frequencies (once or three times) on the volatile profile of Sichuan-style chili oil. The flavor substances of chili oil were analyzed using a combination of headspace gas chromatography–ion mobility spectrometry (HS-GC-IMS), headspace gas chromatography–mass spectrometry (HS-GC-MS), and electronic nose (E-nose) investigations. The variable importance in projection (VIP) values were calculated using partial least squares discriminant analysis (PLS-DA), and the key flavor substances in chili oil were screened by calculating the relative odor activity (ROVA) value. With lipidomics, the correlation between key differential lipids and volatile flavor substances in chili oil was investigated. The effects of different oil temperatures and splashing frequencies on the characteristic aroma of chili oil were clarified. This study provides informative chemical insights into the aroma profile of Sichuan-style chili oil affected by the splashing oil temperature and frequency.

## 2. Materials and Methods

### 2.1. Main Materials and Reagents

Erjingtiao dried chili pepper (*Capsicum annuum* L. *‘Erjingtiao’*) was purchased from the Longquanyi district vegetable wholesale market, Chengdu, Sichuan Province. Third-grade rapeseed oil (Yihai Kerry, Chongqing, China) was purchased from supermarkets in the Longquanyi district, Chengdu, Sichuan Province.

MS-grade methanol, MS-grade acetonitrile, and HPLC-grade 2-propanol were purchased from Thermo Fisher Scientific, Inc. (Cleveland, OH, USA), HPLC-grade formic acid and ammonium formate were purchased from Sigma-Aldrich (St. Louis, MO, USA), and 2-methyl-3-heptanone was purchased from Beijing Manhag Biotechnology Co., Ltd. (Beijing, China).

### 2.2. Sample Preparation

Chili oil was prepared according to previous studies with minor modifications [5,32]. Erjingtiao dried red pepper was ground using a grinding machine through a 50-mesh sieve and divided into four parts. The chili was divided into portions weighing 7 g. Next, 200 g of rapeseed oil was weighed and heated to 215 °C before it was allowed to cool to 210 °C, 180 °C, and 150 °C, respectively. Exactly 30 mL of oil was poured over a portion of chili powder as the oil temperature came to 210 °C, 180 °C, and 150 °C for each oil splashing. For triple oil splashing, the oil was poured into a portion of chili powder three times (10 mL each time). The 10 mL splashing was performed as the oil temperature came to 210 °C, 180 °C, and 150 °C, and the chili oil sample was continuously stirred using a magnet stirring machine (Bkmamlab, Changde, Hunan, China) for 1 min during the intervals of oil temperature dropping. All four sample preparations were performed in triplicate. The chili oil sample was allowed to cool naturally and left in the dark covered with a lid for 24 h for flavor extraction. The chilled oil supernatants were collected for analysis.

### 2.3. GC-IMS Analysis of Volatile Compounds in Chili Oil

The HS-GC-IMS method was based on a previously published study [33] with slight modifications. For GC–IMS analysis, 1 g of chili oil was incubated in a 20 mL headspace bottle at 80 °C for 20 min.

#### 2.3.1. GC Conditions for GC-IMS

A Flavor Spec type GC-IMS instrument (G.A.S. GmbH, Frankfurt am Main, Germany) was used with an MXT-5 chromatographic column (15 m × 0.53 mm × 1 μm, RESTEK, PA, USA). The column temperature was 60 °C, the carrier gas was nitrogen with purity of ≥99.999%, the IMS temperature was set to 45 °C, and the carrier gas flow rate was 2 mL/min from 0 to 10 min, increased to 10 mL/min from 10 to 20 min and 100 mL/min from 20 to 40 min.

#### 2.3.2. IMS Conditions

The drift tube length was 9.8 cm, with a linear voltage of 500 V/cm within the tube. The drift tube temperature was 45 °C, the drift gas was nitrogen with purity of ≥99.999%, and the drift gas flow rate was 150 mL/min. Qualitative analysis was conducted by referencing the built-in NIST and IMS databases (G.A.S., Dortmund, Germany, version 0.4.03) of the GC×IMS Library Search based on the retention indices, retention times, and migration times of the volatile organic compounds.

### 2.4. PLS-DA and VIP Value Analysis

PLS-DA is a supervised analysis method that can effectively reduce the complexity of data and judge the differences in the composition of volatile flavor compounds between samples through pre-set classification [34]. The GC-IMS data were built using a PLS-DA model. The prediction parameters of the PLS-DA model demonstrate the predictive ability of the model. Permutation tests were carried out using SIMCA software (version 14.1) to verify the fit of the PLS-DA model and ensure its stability and reliability. The variable important for the projection (VIP value) formed with PLS-DA was used to analyze the key variables [35]. A VIP value > 1.0 indicated that it played an important role in the discriminant process of the model [36].

### 2.5. GC-MS Analysis of Volatile Compounds in Chili Oil

Gas chromatography–mass spectrometry (GC-MS) was performed according to Yu’s method with slight modifications [37]. In total 5 mL of chili oil was transferred into a 20 mL headspace sample vial, and 5 μg/mL of 2-methyl-3-heptanone was added as an internal standard to quantify volatile organic compounds (VOCs).

#### 2.5.1. GC Conditions

The analysis was carried out using an SQ680 GC-MS instrument (PerkinElmer, Inc., Waltham, MA, USA) equipped with an Elite-5MS chromatographic column (30 m × 0.25 mm × 0.25 μm). The carrier gas was 99.999% He at a flow rate of 1.0 mL/min. The initial column temperature was set to 40 °C, maintained for 3 min, then increased to 90 °C at a rate of 5 °C/min, further raised to 230 °C at a rate of 10 °C/min, and held for 7 min.

#### 2.5.2. MS Conditions

An electron impact (EI) ion source with an electron energy of 70 eV was used. The ion source temperature was 230 °C. The mass scan range was set from 45 *m*/*z* to 550 *m*/*z* using a standard tuning file.

#### 2.5.3. Qualitative and Quantitative Analyses

Compounds were identified based on a match factor exceeding 700 (with a maximum of 999) using both forward and reverse matching, referencing the NIST 2011 spectral library, and were confirmed by interpreting the mass spectra. Relative content was calculated using the peak area normalization method.

### 2.6. Calculation of Relative Odor Activity Value

The relative odor activity value (ROAV) is used to identify key volatile flavor compounds in food and to evaluate the contribution of volatile compounds to the overall aroma [38]. The OAV was calculated by comparing the concentration of each volatile compound with its odor threshold (OT). An OAV > 1 meant the compound significantly contributed to the flavor of chili oil. The OAV of volatile compound in the chili oil exceeded 1 and contributed the most to the flavor of chili oil, and the ROAV of this volatile compound was set at 100. The ROAV of the other VOCs was calculated using the following formula:$$ROAV\approx \frac{Tm}{Cm}\times \frac{Ci}{Ti}\times 100$$
where *Cm* and *Tm* are the content (μg/kg) and threshold (μg/kg) of the compounds that contribute most to the overall flavors of chili oil; and *Ci* and *Ti* are the content (μg/kg) and threshold (μg/kg) of each volatile compound.

The volatile compounds of ROAV > 1 were considered key flavor components [39]. Volatile compounds with an ROAV between 0.1 and 1 are thought to modify the overall flavor [40,41].

### 2.7. E-Nose Analysis

The E-nose detection method was slightly modified according to the protocol described in [5]. Two milliliters of chili oil was transferred into a 10 mL headspace vial and sealed for subsequent use. An electronic nose (Alpha MOS Fox 4000, Toulouse, France) was employed for analysis, with the samples incubated in the incubator at 70 °C for 5 min. The injection volume was 2000 μL, with an injection speed of 2000 μL/s, and the samples were injected manually. Each sample was analyzed for 120 s with a data acquisition delay of 150 s.

### 2.8. Determination of Fatty Acids

Fatty acids were detected as described in [42].

#### 2.8.1. Fatty Acid Methyl Esterification Treatment

Chili oil (2 mg) was dissolved in 1.5 mL n-hexane and methylated with 40 μL of methyl acetate and 100 μL of sodium methoxide for 30 min with mixing by vertexing. The supernatant was filtered through a 0.45 μm filter membrane and analyzed by gas chromatography (6890N; Agilent, Santa Clara, CA, USA).

#### 2.8.2. GC Analysis

The injection port temperature was set to 250 °C with no split, and the carrier gas was helium with a column flow rate of 1.8 mL/min. A CP-Sil88 column (CP7489, 100 m × 0.25 mm × 0.2 µm, Agilent) was used. The temperature program was as follows: initial temperature of 45 °C for 5 min; from 45 °C to 175 °C at a rate of 13 °C/min; from 175 °C to 215 °C at a rate of 4 °C/min. The injection volume was 1 μL. Fatty acids were identified by comparing their retention times with those of a standard FAME mixture (#463) (GLC463; NuChek. Prep, MN, USA).

### 2.9. Lipidomics Analysis of Chili Oil

We added 200 μL of water and 20 μL of lipid mixture internal standard (internal standard: SPLASH^®^ LIPIDOMIX MASS SPRC STANDARD, AVANTI, 330707-1EA) to 5 μL of chili oil, then thoroughly vortexed it before adding 800 μL methyl tert-butyl ether (MTBE), and vortexed it again. The mixture was then mixed and vortexed with 240 μL precooled methanol, ultrasonicated in a 10 °C water bath for 20 min, followed by 14,000× *g* centrifugation at 10 °C for 15 min. The supernatant was taken and freeze-dried with liquid nitrogen before it was redissolved with 200 μL 90% isopropanol/acetonitrile solution. The complex solution was fully vortexed, 90 μL of the solution was centrifuged at 14,000× *g* at 10 °C for 15 min, and the obtained supernatant was injected for analysis.

A UHPLC Nexera LC-30A ultraperformance liquid chromatograph (SHIMADZU, Kyoto, Japan) equipped with a C18 column (1.7 μm, 2.1 mm × 100 mm; Waters, Milford, MA, USA) was used for separation. The column temperature was 45 °C, and the mobile phase consisted of the following: A: acetonitrile aqueous solution (acetonitrile: water = 6:4, *v*/*v*), with 0.1% formic acid and 0.1 mM ammonium formate and mobile phase; and B: acetonitrile isopropanol solution (acetonitrile: isopropanol = 1:9, *v*/*v*) with 0.1% formic acid and 0.1 mM ammonium formate. The gradient elution procedure was 30% solvent B for 0–2 min; 30%-100% solvent B for 2–25 min; and 30% solvent B for 25–35 min.

The analytes were detected by electrospray ionization (ESI) in positive and negative ion modes. Ten fragments (MS2 scan, HCD scans) were collected after each full scan. The resolution of MS^1^ was 70,000 at *m*/*z* 200, and that of MS^2^ was 1750 at *m*/*z* 200. The heater temperature was set to 300 °C, sheath gas flow rate to 45 arb, aux gas flow rate to 15 arb, sweep gas flow rate to 1arb, spray voltage to 3.0 kV, and capillary temperature to 350 °C. The non-targeted lipidomics data processing in this study referred to Alseekh’s study [43].

### 2.10. Kyoto Encyclopaedia of Genes and Genomes (KEGG) Analysis of Differential Lipids

Identified differential lipids were annotated using the KEGG Compound Database (http://www.kegg.jp/kegg/compound/, accessed on 2 November 2024), and the annotated differential lipids were mapped to the KEGG Pathway Database (http://www.kegg.jp/kegg/pathway.html, accessed on 3 November 2024) for enrichment analysis.

### 2.11. Statistical Analysis

The significance of the differences and standard deviation of the ROAV were determined using SPSS (version 25.0; IBM Corporation, Armonk, NY, USA), with the cut-off for significance offset to *p* < 0.05. Radar plots were generated using Origin 2024 (Version 18.1, Origin Lab Corporation, Northampton, MA, USA), and PLS-DA was performed using SIMCA software (Version 18.1, Umetrics, Umeå, Sweden). Fingerprints were drawn using the Reporter plug-in (FlavourSpec; G.A.S., Germany). The Pearson correlation test and correlation heat map were analyzed online with Knowall (https://cnsknowall.com, accessed on 15 November 2024). Lipid metabolic pathways were analyzed using the MetaboAnalyst platform (https://www.metaboanalyst.ca, accessed on 16 November 2024).

## 3. Results and Discussion

### 3.1. Volatile Profile of Chili Oil Identified by GC-IMS

GC-IMS was used to characterize the VOCs in the chili oil samples at different oil temperatures and splashing frequencies, identifying 63 volatile compounds (Table 1), including 14 alcohols, 2 acids, 19 aldehydes, 8 ketones, 9 esters, 7 olefins, and 4 others.

Figure 1A shows the volatile fingerprint of chili oil and reflects the variation in volatile compounds in different chili oil samples. The fingerprints of chili oil were divided into three zones: I, II, and III.

In zone I, the concentrations of butyl acetate, 2-methylpyrazine, 1-octaldehyde, 2-hexanol, cyclohexanone, cyclopentanone, 1-octen-3-one, and other volatile compounds in the 210 °C group were significantly higher than those in the other three chili oil groups. Among these, octanal is mainly derived from oleic acid oxidation, which is characterized by a fatty aroma [44,45]. 2-methylpyrazine can be formed through the condensation of two α-amino-ketones through Maillard reaction, which confers a nutty and barbecue aroma [45,46]. 1-octen-3-one is mainly characterized by a mushroom-like aroma [47]. Butyl acetate, cyclohexanone, and cyclopentanone were not produced when chili oil was poured onto Erjingtiao pepper at 180 °C [5], and researchers found an increase in cyclohexanone in roasted chicken marinated with tea at 240 °C [48]. Octanal, 1-octen-3-one, and 2-methylpyrazine were produced in large quantities during frying Youtiao at 190 °C and roasting beef at 200 °C [49,50]. These results show that butyl acetate, cyclohexanone, cyclopentanone, octanal, 1-octen-3-one, and 2-methylpyrazine were more likely to be formed at high temperatures above 180 °C. It was suggested that 210 °C oil splashing could promote the retention of pyrazines, ketones, and butyl acetate in finished red oil.

In zone II, the concentrations of 3-methyl-2-butenal, p-cymene, β-ocimene, (+)-limonene in 210 °C and triple-splashing groups were significantly lower than those in the 150 °C and 180 °C groups. Limonene, with caramel, citrus, and other aromas, had its highest content in 150 °C chili oil. The formation and retention of olefins in 180 °C and 150 °C chili oil were more concentrated. It was previously reported that the free radical reaction and double-bond position isomerization of linoleic acid and linolenic acid could facilitate the formation of olefins at 180 °C [51]. In zone III, the concentrations of (E)-2-heptenal, heptanal, butyraldehyde, and 3-methylbutyraldehyde in 180 °C and 150 °C samples were significantly higher than those in 210 °C samples. The above-mentioned aldehydes, which feature a fatty aroma, may be formed via linoleic acid oxidation [52]. The relative content of most aldehydes in chili oil reportedly could reach their highest levels when the splashing temperature was in the range of 160–180 °C [53].

According to the relative content stacking diagram in Figure 1B and the peak area in Figure 1C of various compounds in chili oil, the oil splashing temperature had a significant effect on the profile of volatile components, especially aldehydes and ketones. Aldehydes were the main volatile substances in all chili oil samples. The relative contents of aldehydes at 180 °C were higher than those of other types of compounds. According to Table 1, the contents of ketones were the highest in 150 °C chili oil samples; to be more specific, the concentration of 1-penten-3-one in 150 °C samples was significantly higher than the other three chili oil samples. Studies have shown that ketones are generated via the oxidative decomposition of unsaturated fatty acids [54].

The triple-splashing samples underwent three consecutive oil splashing processes at 210 °C, 180 °C, and 150 °C. The VOCs generated at 210 °C would become the substrates for the subsequent 180 °C and 150 °C splashing. In zone I, cyclohexanone, cyclopentanone, 1-octanal, 1-octen-3-one, and 2-methylpyrazine were the highest in the 210 °C group, while their contents decreased in the triple-splashing group. Different heating temperatures can lead to different lipid oxidation, amino acid degradation, or Maillard reaction products, which alter the composition of different volatile substances as the degree of heat treatment increases [55]. The Maillard reaction usually occurs above 154 °C and is accompanied by Amadori rearrangements to form aldehydes, furans, ketones, and acids [56]. Maillard reaction products could further interact with Strecker-degraded oxidized α-amino acids to form 3-methylbutyraldehyde, benzaldehyde, aminoketones, and heterocyclic amines. These substances interact with other lipid oxidation products to form pyrazines, thiazoles, and thiols [55,57,58]. The decrease in cyclohexanone, cyclopentanone, octanal, and 1-octen-3-one in the triple-splashing samples indicated that these substances continued to react with the Strecker-degraded oxidation products at 180 °C. The contents of β-ocimene in zone II and 1-pentanol in zone III were reduced in the triple-splashing samples. This decrease is due to the volatilization and dissipation of alcohols and the decomposition of olefins under the intense conditions of repeated frying [51].

### 3.2. Volatile Profile of Chili Oil Identified by GC-MS

A total of 62 VOCs were identified, including 6 alcohols, 20 aldehydes, 9 ketones, 4 esters, 14 olefins, 2 ethers, 3 acids, and 4 others (Table 2). There were 57 volatile compounds in the 210 °C chili oil sample and 33 volatile compounds in the 180 °C chili oil samples. Aldehydes accounted for the highest proportion of all the volatile compounds in chili oil. Lipids are oxidized and decomposed during high-temperature cooking to produce more aldehydes and ketones [52]. In agreement with the GC-IMS results, aldehydes and ketones accounted for the largest proportion in all chili oil GC-MS profiles, and the aldehyde, ketone, and olefin contents in the 210 °C chili oil samples were significantly higher than those in other chili oils (Table 2), and the contents of aldehydes in the chili oil gradually decreased at 180 °C and 150 °C. The triple-splashing samples produced the most olefins, and the olefin contents increased with the increase in oil splashing temperature; olefin contents also increased significantly after roasting buckwheat at 210 °C compared to at 180 °C [55].

The MXT-5 polar column for GC-IMS and the Eite-5MS non-polar column for GC-MS detected 63 and 62 volatile compounds, respectively. Although the number of compounds was similar, there were significant differences. According to Table 1 and Table 2, 49 volatile compounds, such as (E, E)-2,4-decadienal and benzaldehyde, were unique to the GC-IMS results. Meanwhile, 48 volatile compounds, including methylpyrazine and tetrahydrofuran, were unique to GC-MS. Fourteen volatile compounds, mainly aldehydes, olefins, alcohols, and esters, were detected in both the non-polar and polar columns. The compound classes were similar to those detected in both non-polar and polar columns for salmon VOCs [45]. Non-polar columns are primarily separated based on the boiling point of the analyte [45]. The compounds detected using Eite-5MS as non-polar columns were aldehydes (20), alcohols (6), and ketones (9). (E, E)-2,4-decadienal and benzaldehyde were separated in the HP-5MS non-polar column [15]. The polar groups of the polar column stationary phase were separated by dipole–dipole and hydrogen bonding interactions with the analytes [59]. The compounds detected with the MXT-5 as a polar column were olefins (7), aldehydes (19), esters (9), and acids (2). Methylpyrazine and tetrahydrofuran were also separated using a TG-WAX polar column [5]. GC-MS is widely used for the analysis of volatile compounds because of its precision and versatility [60]. Compared to GC-MS, GC-IMS is a highly sensitive method for the detection of trace organic compounds [61] and can detect dimers or polymers [55]. Owing to the complexity of volatile compounds in chili oil, the use of two columns with different polarities provided a complete and adequate understanding of the volatile profiles of chili oil. Finally, 111 compounds were identified using GC-IMS and GC-MS.

### 3.3. E-Nose Analysis

An e-nose can quickly differentiate the overall volatile characteristics of a sample, and each of its sensors is designed to capture specific categories of volatile substances [62]. Figure 2A shows the electronic nose signal radar map of the chili oils. The 12 sensors in the response array showed various response intensities, indicating that different oil temperatures and oil splash frequencies significantly affected the overall aroma of chili oil. The P30/2, P30/1, PA/2, and P40/2 sensors with high response values (Figure 2A) were sensitive to ketones, alcohols, aldehydes, and organic compounds [63]. The LY2/LG, LY2/G, LY2/AA, LY2/Gh, LY2/gCT1, and LY2/gCT sensors were sensitive to sulfides, methylamines, organic compounds (ethanol, acetone, ammonia), anilines, amine compounds, and butane or propane [63], yet all chili oils showed very low responses in these sensors. The E-nose signals were consistent with those of the GC-MS and GC-IMS analyses. The E-nose analysis could characterize chili oils much faster than GC, albeit with less detail. The response intensity of the E-nose sensors could be a great evaluation tool for rapidly monitoring the intensity of certain signature flavors of chili oil.

PCA is an effective tool for reducing the dimensions of the E-nose data and can be used to identify potential correlations among multiple variables [64]. As shown in Figure 2B, the first (83.6%) and second principal components (6.0%) accounted for approximately 90% of the total variance. The volatile flavor compounds in chili oil changed because of the changes in oil splash temperature. Figure 2C shows the correlation of each sensor with each chili oil sample; the greater the variable load, the stronger the correlation. The 210 °C and 180 °C chili oil aggregated on the positive axis of PC1, mainly associated with the P30/2, P30/1, PA/2, and P40/2 sensors, while the 150 °C and triple chili oil aggregated on the negative axis of PC1, mainly associated with the LY2/Gh, LY2/G, LY2/gCT1, LY2/gCT1, and LY2/gCT sensors. However, the four types of chili oils overlapped with each other, as shown in Figure 2B,C, indicating that there was no significant difference in the types of volatile flavor compounds. The effect of chili oils prepared at different oil temperatures on the types and contents of VOCs was not sufficient to distinguish them using the E-nose, and it was necessary to further analyze the VOCs by combining GC-MS and GC-IMS.

### 3.4. Key Volatile Flavor Compounds of Chili Oil Analysis

#### 3.4.1. VIP Analysis

The variable importance in prediction (VIP) is an important index for evaluating the significance of variables in distinguishing samples and can reflect the contribution of flavor substances. Compounds with a VIP > 1 are usually considered to contribute significantly to the flavor profile [65]. VIP was used to screen for key flavor compounds, as it is generally recognized that the greater a compound’s VIP value, the greater its flavor contribution [36]. Through PLS-DA, 17 key flavor compounds with VIP > 1 were screened out from the chili oil samples (Figure 3), including (E)-2-heptenal, 2-methyl-3-ketotetrahydrofuran, 2-octanone, (+)-limonene, β-ocimene, p-cymene, heptaldehyde, 3-methyl-3-buten-1-ol, 2-pentanol, methyl pentanoate, 2-methyl-1-propanol, ethanol, 1,1-diethoxy ethane, tetrahydrofuran, acrolein, ethyl formate, and 2-methyl butanal. β-ocimene, p-cymene, 2-methyl-3-ketotetrahydrofuran, heptaldehyde, and (E)-2-heptenal were the top five key flavor substances in terms of VIP value. There were no significant differences in the content of β-ocimene, which has citrusy, woody, and green aromas, in the 210 °C, 180 °C, and 150 °C chili oil samples; however, its content in the triple-splashing sample was significantly lower than in them. The distribution of p-cymene, with woody and lavender-like aromas, was consistent with that of β-ocimene. The results indicate that the temperature of oil splashing did not significantly affect the formation of β-ocimene and p-cymene, while repeated oil splashing would lead to the decomposition of β-ocimene and p-cymene. Heptaldehyde and (E)-2-heptenal with fatty aroma were significantly higher than the other three chili oil samples at 180 °C. The VIP of heptanal, (E)-2-heptenal, pentanol, methyl butyraldehyde, 2-methyl-1-propanol, ethanol, and acrolein in chili oil made from Erjingtiao at 180 °C exceeded 1, meaning they were also considered to be key flavor substances in chili oil [33]. β-ocimene and p-cymene were considered to be key flavor compounds with VIP > 1 in this study, but they were not detected in chili oil made from Erjingtiao at 180 °C, because while β-ocimene and p-cymene can be separated and identified with MXT-5 or DB-5MS columns due to their weak polarity [15], they cannot be separated and identified using WAX or DB-FFA strong-polarity columns, resulting in them remaining undetected [33,66].

#### 3.4.2. Analysis of ROAV

The contribution of volatile compounds to the overall aroma characteristics of chili oil could not be reflected only by their relative content, because the intensity of the odor was closely related to the threshold and the content of the volatile compounds. The ROAV has been widely used to identify key volatile flavor compounds in food and to evaluate the contribution of volatile compounds to the overall aroma [38,67]. The threshold values and odor descriptors of various compounds were sourced from a book named “*Odor Threshold Compilations of Odor Threshold Values in Air, Water, and Other Media (second enlarged and revised edition)*”, as well as from a study by Liu Feifei [68], which examined the flavor characteristics of oil systems. Volatile compounds with an ROAV > 1 are generally considered key flavor components and contribute significantly to the flavor. There were 19 volatile compounds with an ROAV > 1 in chili oils, with 13, 11, 11, and 10 substances in 210 °C, 180 °C, 150 °C, and triple-splashing chili oil samples, respectively. The ROAV of 2-methylbutyraldehyde was 100 (Table 3), making it the largest contributor to the flavor at 210 °C and 180 °C, contributing malty aroma to chili oil. The ROAV of β-ocimene was 100 (Table 3); it was the largest contributor to the flavor at 150 °C, conferring citrusy, woody, and green aromas to the chili oil. The ROAV of limonene was 100 (Table 3), which contributed lemon and orange aromas, and was the largest contributor to the flavor of triple-splashing chili oi. The OAV of 2,4-nonadienal, benzaldehyde, β-ocimene, p-cymene, α-pinene, caryophyllene, and limonene in chili oil at 150 °C exceeded 1, which means they were also considered to be key flavor substances in chili oil [15]. Further, the ROAV of 1-octen-3-ol, 2-methylbutyraldehyde, octanal, and (E)-2-octenal in chili oil made from Erjingtiao at 180 °C exceeded 1, which means they were also considered to be key flavor substances in chili oil [5]. The OAV of furfural, trans-2-decenal, and nonanal in chili oil made from Erjingtiao at 230 °C exceeded 1, so they were also considered to be key flavor substances in chili oil [66]. (E, E)-2,4-heptadienal was exclusively produced under high-temperature oil splashing at 210 °C (Table 3). Consequently, it was not identified in chili oil derived from oil splashing at 180 °C using Erjingtiao [5].

#### 3.4.3. Identification of Key Volatile Flavor Compounds by VIP and ROAV

According to the Venn diagram in Figure 4A, GC-IMS and GC-MS detected 111 volatile compounds in total, of which only 14 were found by both methods, suggesting that the volatile compounds separated and identified using different chromatographic columns can complement each other. According to VIP > 1 and ROAV > 1, a total of 31 key volatile flavor substances were screened; only 5 VOCs, namely 2-methyl butanal, (E)-2-heptenal, β-ocimene, p-cymene, and limonene, were identified by both methods. These five key aroma compounds were weakly polar and could be detected in both the polar and non-polar columns. (E, E)-2,4-heptadienal, furfural,2,4-nonadienal, nonanal, 2,4-decadienal, benzaldehyde, benzeneacetaldehyde, trans-2-decenal, α-pinene, caryophyllene, α-phellandrene, 1-butanol, and α-terpinolene were isolated and detected using the Elite-5MS non-polar column of GC-MS due to their boiling point and weak polarity [15]. 2-methyl-3-ketotetrahydrofuran, 2-octanone, heptaldehyde, 3-methyl-3-buten-1-ol, 2-pentanol, methyl pentanoate, 2-methyl-1-propanol, ethanol, 1,1-diethoxy ethane, tetrahydrofuran, acrolein, and ethyl formate were detected using the MXT-5 polarity column of GC-IMS because of their polarity [69]. A study found that the concentration of primary terpenes, including α-pinene and α-terpinolene, diminishes during the heating of rapeseed oil [70]. In rapeseed oil-based chili oil, compounds such as α-phellandrene, β-ocimene, and limonene were identified as essential flavor components, indicating that these terpenes are not generated by heating rapeseed oil but are unique to chili oil. As shown in Figure 4A, ethanol, heptanal, and α-terpinolene were detected by both IMS and MS, but the ROAV showed that ethanol and heptanal were not the key aroma components of chili oil, while the VIP value showed that α-terpinolene was not a key aroma component of chili oil. Heptanal had an OAV > 1 in chili oil made with Erjingtiao at 230 °C, meaning it was considered a key aroma substance, showing citrus and fatty aromas [66]. Ethanol in the chili oil made with Erjingtiao at 180 °C was considered a key aroma substance, showing alcohol aroma [69]. α-terpinolene is a key volatile flavor substance in pepper (*Zanthoxylum bungeanum* Maxim.) [71] and is mainly derived from spices in chili oil. It would be necessary to further determine whether the 31 aroma compounds are key aromas by omitting the recombination experiments.

As shown in Figure 4B, 210 °C chili oil contained the most key flavor components. We noted that at 210 °C, which is the smoking point of rapeseed oil, oil components might become more reactive as they approach the smoking point, which means they could promote the interaction between oil and chili powders, thereby forming the signature chili oil aroma. Cluster analysis suggested that the 210 °C sample was significantly different from oils prepared in other procedures. The common key volatile flavor compound identified through both the VIP and ROAV, 2-methyl butanal, had its highest content at 210 °C. As shown in Figure 1B, the contents of most key aldehydes (4-heptadienal, trans-2-decenal, 1-butanol, α-pinene, caryophyllene, and α-phellandrene) and 1,1-diethoxy ethane were highest in 210 °C chili oil, indicating that these fatty, spicy, and grassy aroma [69] compounds need to be generated at high temperatures at around 210 °C.

The 180 °C chili oil was abundant with key flavor components. The contents of common key volatile flavor compounds, (E)-2-heptenal, β-ocimene, and p-cymene, were at their highest at 180 °C. The contents of nonanal, 2,4-decadienal, and α-terpinene were the highest in 180 °C samples. Nonanal and 2,4-decadienal provide chili oil with fatty and citrusy aromas [66]. α-terpinene conferred a pine aroma to the chili oil [71]. 2-octanone, heptaldehyde, and ethyl formate were the most abundant in 180 °C samples, indicating that 180 °C was conducive to the formation or retention of these substances.

The common key volatile flavor compound limonene was most abundant at 150 °C. Furfural and benzaldehyde were the highest in the 150 °C samples, which provided bread-like, nutty, and grass aromas for chili oil. As shown in Figure 4B, cluster analysis suggested that triple splashing samples and 150 °C samples were most similar in terms of key flavors. The flavor compounds of chili oil were mainly composed of Maillard reaction [25], lipid primary oxidation [26], and lipid secondary oxidation products [72] during the 24 h extraction process. The final flavor of chili oil may depend more on the temperature of the extracted oil than on the temperature of the poured oil. Since 150 °C is lower than the temperature required for the Maillard reaction [58], triple splashing had the advantage of increasing the volatile flavor compounds produced by the Maillard reaction compared to 150 °C, such as 2-methylpyrazine and 2-methylbutyraldehyde in Table 1 and heptanal, phenylacetaldehyde, nonanal, 2-hexenal, 2-butanone, and 2-propylfuran in Table 2. Compared with the 210 °C samples, the triple-splashing samples had the advantage of retaining more olefins, such as limonene and β-ocimene, as shown in Table 2, increasing the citrusy and grassy aromas of chili oil. The disadvantage was that the triple-splashing samples showed reduced volatile flavor compound contents, resulting in a diminished overall flavor intensity relative to the 210 °C samples. At 180 °C, the Maillard reaction products, retention of olefins, and overall flavor intensity of traditional triple chili oil were enhanced. Given the premise of ensuring the flavor of Sichuan-style chili oil, simplifying the industrialization process of Sichuan-style chili oil using 180 °C oil splashing is undoubtedly the best choice.

### 3.5. Free Fatty Acid and Lipidomics Analyses of Chili Oil

The thermal oxidation of fatty acids can form volatile compounds such as acids, esters, furans, aldehydes, alcohols, and ketones [73]. The principal fatty acids in chili oil are oleic acid (C18:1n9c), linoleic acid (C18:2n6), and α-linolenic acid (C18:3n3) [74]. In analyzing free fatty acids in different chili oil samples, it was also found that the main free fatty acids of chili oil were palmitic acid (C16:0), stearic acid (C18:0), oleic acid (C18:1n9c), linoleic acid (C18:2n6), α-linolenic acid (18:3n-3), and palmitoleic acid (C16:1n7). As shown in Figure 5A, linoleic acid and α-linolenic acid showed significant differences among the different chili oil samples. Linoleic acid can be further oxidized to hexanal and heptanal at high temperatures [75]. The contents of hexanal and heptanal were the highest in 210 °C and 180 °C samples, as shown in Table 2. It was highly likely that linoleic acid was oxidized at high temperatures, as the content of α-linolenic acid decreased significantly when it was heated above 180 °C [76]. The content of α-linolenic acid in 210 °C samples was also significantly lower than that in other samples. Therefore, the formation of VOCs in chili oil was closely related to the hydrolysis and oxidation of free fatty acids, especially linoleic acid and α-linolenic acid.

Non-targeted lipidomics was used to analyze the composition of chili oil produced at different oil temperatures and splash frequencies. As shown in Figure 5B, 1908 lipid molecules from 40 subclasses were identified in both the positive and negative ion modes. Triglycerides (TGs) made up the most abundant lipid subclass. Free fatty acids formed by TG decomposition play a crucial role in flavor formation [77]. Based on fold change (FC) analysis and *t* tests/non-parametric tests (*p* < 0.05), we conducted an analysis of the differences in all detected lipid molecules. A total of 411 different lipids were identified, and the analysis results are displayed in the form of volcano maps. The results are presented in Figure 5C. Differential lipid molecules that met *p* < 0.05 are represented with different colors. The smoke point of rapeseed oil was reached at 210 °C. Accelerated lipid decomposition at the smoke point produces more free fatty acids [78]. The lipid composition of the 210 °C samples was the most different from that of 150 °C samples, which reflects that the lipid decomposition process of rapeseed oil was accelerated at the smoke point temperature, making it more conducive to forming volatile compounds. These findings indicate that the notable variation in the production of VOCs observed between 210 °C and other oil splash temperatures, as illustrated in Figure 4B cluster analysis, can be attributed to the differing degrees of lipid decomposition at these various temperatures, and the oxidative pyrolysis of lipids could be an important generation pathway of VOCs in chili oil.

To explore the key metabolic pathways of lipids in chili oil, a Kyoto Encyclopedia of Genes and Genomes (KEGG) pathway enrichment analysis was performed based on the differential lipids in different chili oil samples. Figure 5D shows that 411 differentially expressed lipids were significantly enriched in the three KEGG pathways (*p* < 0.05). The glycerolipid metabolic pathway (map00561), glycerophospholipid metabolic pathway (map00564), and sphingolipid metabolic pathway (map00600) were enriched. Glycerolipid metabolism is the key metabolic pathway for the production of flavor substances in heat-treated grass carp [73]. Therefore, glycerolipid metabolism may be related to the formation of key volatile flavor substances in chili oil. In addition, glycerophospholipid and sphingolipid pathways might also play roles in flavor development. Pan [79] showed that glycerophospholipid metabolism was the main pathway of lipid oxidation in hazelnut oil. The unsaturated fatty acids generated from glycerophospholipid metabolism were reported to serve as crucial precursors for the synthesis of aromatic compounds [80]. Li [81] discovered that sphingolipid metabolism was the most significantly enriched lipid metabolic pathway during the storage oxidation of sea buckthorn fruit oil. Additionally, the decrease in sphingolipid metabolites was observed in oil during high-temperature or frying processes [81]. Yu [82] explored the potential link between the lipids and flavors of watermelon seeds and identified sphingolipid metabolism as the major metabolic pathway, as indicated by KEGG metabolic pathway enrichment analysis. Key differential lipids were selected from each group based on FC > 1 and VIP > 1. A total of 18 triglyceride subclasses (TGs), key differential lipids were screened. Free fatty acids, which are the main oxidation products of lipids, can be enzymatically or non-enzymatically converted into free radicals upon heating [83]. Free radicals can alter the structure of TGs, facilitating their decomposition and oxidization [83,84]. Chili oil contains 18 key differential TGs; therefore, the key flavor substances of chili oil may mainly originate from the oxidative pyrolysis TGs. The 210 °C group contained the most types of and most abundant TG content, and the contents of volatile flavor substances such as aldehydes, ketones, and esters in the 210 °C chili oil samples were significantly higher than those in the other chili oil samples in Table 2. Therefore, TG plays an important role in the formation of aromatic compounds in chili oil.

### 3.6. Correlation Analysis of Differential Lipids with Key Flavor Substances

The network diagram in Figure 6A shows that 18 key TG lipids were closely related to glycerolipid metabolism. Research has found that lipid oxidation can produce free fatty acids, which through enzymatic or non-enzymatic reactions generated radicals at higher temperatures [83]. These radicals could alter the structure of triacylglycerols (TGs), making them more susceptible to decomposition and oxidation [83,84]. TGs played a significant role in the formation of aromatic compounds [30]. Compared to the key differential lipids in the triple splashing chili oil, TG (18:3_13:0_18:2), TG (18:1_14:3_18:1), TG (18:1_13:0_18:3), and TG (16:0_11:1_20:2) were significantly upregulated, while TG (20:5_22:5_22:6) was significantly downregulated in the 210 °C chili oil. However, there was no significant change in the 150 °C chili oil compared to the triple-splashing chili oil. This confirms the similarity in flavor composition between the 150 °C chili oil and the triple-splashing chili oil, further indicating that the key differential lipids of triacylglycerol subclasses (TGs) were the main precursors of flavor substances in chili oil. Among the 18 key differential lipids of the triacylglycerol subclasses (TGs), the main esterified fatty acids were oleic acid (C18:1), linoleic acid (C18:2), linolenic acid (C18:3), and polyunsaturated fatty acids. The oxidation of oleic acid, linoleic acid, and α-linolenic acid leads to the formation of key volatile compounds in chili oil, such as nonanal, 2,4-decadienal, and (E,E)-2,4-heptadienal [85]. The correlation analysis of key differential lipids and key flavor substances in chili oil is shown in Figure 6B. GC-IMS and GC-MS identified common key volatile flavor substances: 2-Methyl butanal, (E)-2-heptenal, β-Ocimene, p-cymene, and limonene. Among them, (E)-2-heptenal was significantly positively correlated with TG (16:0_11:1_20:2), TG (18:1_13:0_18:3), TG (18:1_14:3_18:1), and TG (18:3_13:0_18:3), and 2-methyl-butanal was significantly positively correlated with TG (16:0_11:1_20:2) and TG (18:1_13:3) (*p* < 0.05).

GC-MS identified unique key volatile flavor compounds: 1,1-diethoxy ethane, D-limonene, 1-butanol, 2,4-nonadienal, and TG (16:0_11:1_20:2) were significantly and positively correlated (*p* < 0.05). TG (18:3_13:0_18:2) was significantly and positively correlated with the linoleic acid content (*p* < 0.05). Tetrahydrofuran and limonene identified by GC-IMS were significantly and positively correlated with TG (18:4_22:3_23:1) (*p* < 0.05).

Studies have shown that the linear aldehydes of C5-C10 are mainly derived from oleic acid, linoleic acid, linolenic acid, and arachidonic acid [27]. Limonene, (E)-2-heptenal, and tetrahydrofuran were significantly positively correlated with α-linolenic acid (*p* < 0.05). 2-methyl butanal, nonanal, 1,1-diethoxy ethane, 1-butanol, and caryophyllene were significantly positively correlated with linoleic acid (*p* < 0.05). Benzene acetaldehyde was significantly and positively correlated with oleic acid (*p* < 0.05). Additionally, TG (18:3_13:0_18:2) showed a significant positive correlation with linoleic acid content (*p* < 0.05). Linoleic acid generates 9(S)-HPODE hydroperoxides, which can further convert to (E)-2-nonenal and 2,4-decadienal [73]. Furthermore, 2,4-decadienal was significantly positively correlated with TG (18:3_13:0_18:2) (*p* < 0.05). Therefore, ω-3 polyunsaturated fatty acids, linoleic acid, oleic acid, and α-linolenic acid in the TG structure were significantly correlated with the key flavor substances of (E)-2-heptenal, 2-methylbutanal, limonene, (E, E)-2,4-heptadienal, 2,4-nonadienal and 2,4-decadienal, which means they may be main contributors to the production of key flavor substances in chili oil.

## 4. Conclusions

This study explored the effects of different oil temperatures and splashing times on the key aromas of chili oils. We found 31 key aroma compounds in the different chili oil groups, and five key aromas in all treatment groups. Oil splashing at 210 °C promoted the formation of esters, ketones, pyrazines, alcohols, and aldehydes, enhancing its fatty, floral, fruity, nutty, and grilled aromas. The three-stage oil splashing (210 °C-180 °C-150 °C) was not conducive to the retention of aldehydes such as (E)-2-heptenal, β-ocimene, and p-cymene, while limonene content increased. Three-stage cooling oil splashing reduced the woody, grassy, and fatty aromas and increased the citrus aroma. Mild oil splashing at 150 °C was beneficial for the production of furanones, olefins, and aldehydes, and provided caramel and citrus aromas. Lipidomics analysis revealed 411 differential lipids in chili oil samples, while KEGG enrichment analysis identified glycerolipid metabolism as the key metabolic pathway in chili oil after oil splashing. This study clarified the role of different oil temperatures (210 °C, 180 °C, 150 °C) on the regulation of key flavor substances in chili oil and provided temperature control standards for industrial production. By adjusting the splashing temperature or adopting a staged temperature treatment, consumers and industry professionals may customize chili oil products with different aroma profiles.

Although our findings could be generalizable to other chili varieties with similar sugars, proteins, and fats profiles, the whole picture of chili oil flavor development would be more comprehensively presented if the effect of other oil types were investigated. Future works may further reveal the molecular mechanisms of specific lipid degradation pathways (e.g., oxidation, enzymatic degradation) and verify the generation route of the key aroma compounds.

## Figures and Tables

**Figure 1 foods-14-01006-f001:**
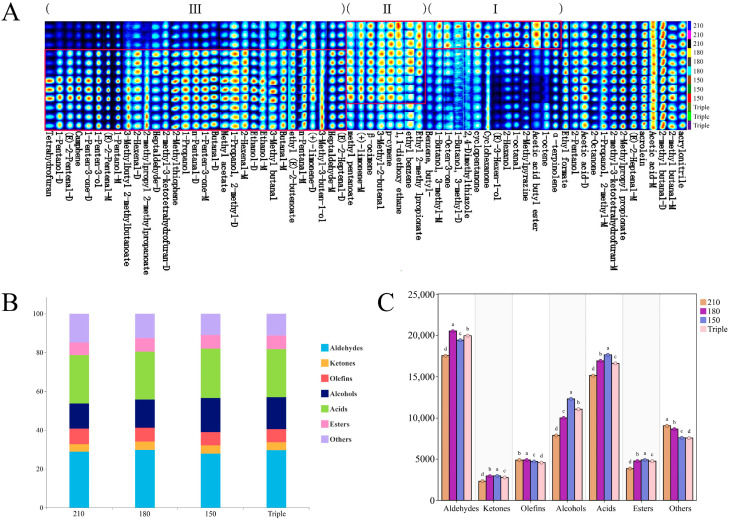
Gas chromatography–ion mobility spectrometry (GC-IMS) analysis of chili oil with different oil splashing temperatures and frequencies. (**A**) Fingerprint of volatile compounds (zone I: volatile organic compounds that were more abundant in 210 °C chili oil than in other chili oils; zone II: volatile organic compounds that were less abundant in triple chili oil than in other chili oils; zone III: volatile organic compounds that were less abundant in 210 °C chili oil than in other chili oils). (**B**) Stacking bar chart of volatile organic compounds (VOCs); (**C**) peak area chart of VOCs in four chili oil samples. Bars with different letters represent significant difference (*p* < 0.05).

**Figure 2 foods-14-01006-f002:**
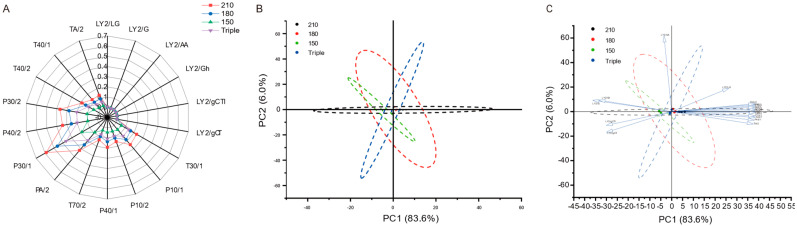
E-nose analysis of chili oil. (**A**) Radar chart; (**B**) PCA score plot; (**C**) Loading plot based on E-nose analysis.

**Figure 3 foods-14-01006-f003:**
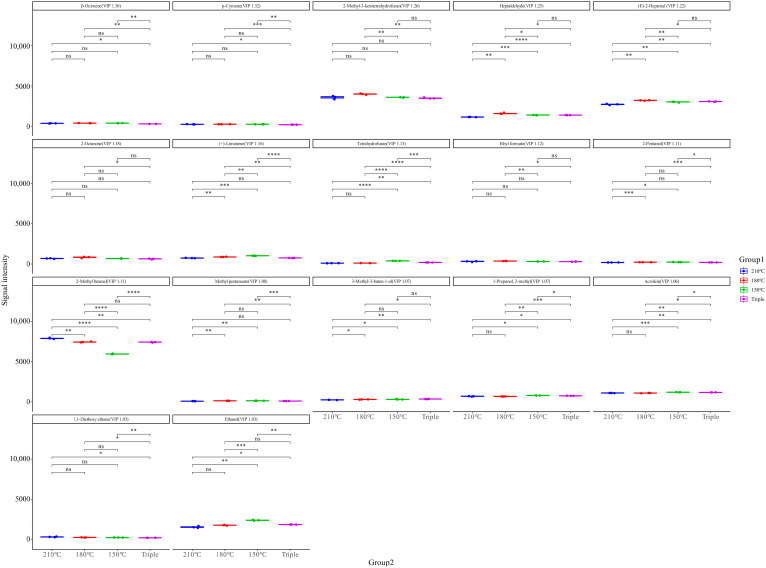
Distribution of 17 key flavor compounds with VIP > 1 of chili oil in the GC-IMS PLS-DA model. * means *p* < 0.05, ** means *p* < 0.01, *** means *p* < 0.001, **** means *p* < 0.0001 and “ns” means no significant difference between the two compared samples.

**Figure 4 foods-14-01006-f004:**
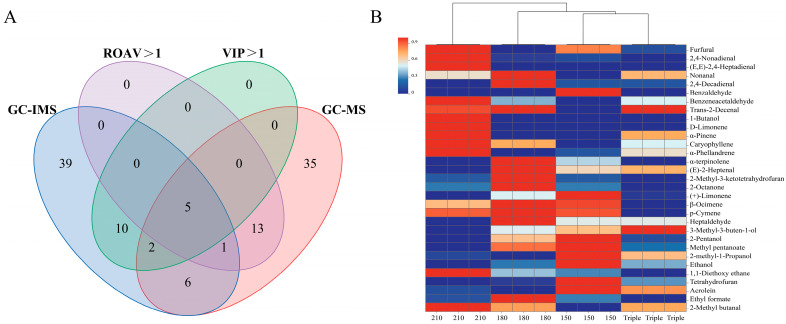
Key volatile flavor compounds of chili oil analysis. (**A**) GC-IMS and GC-MS volatile compound Venn diagrams; (**B**) variable importance in projection (VIP) > 1 and relative odor activity value (ROAV) > 1 volatile compound distribution heat maps of the four chili oil samples.

**Figure 5 foods-14-01006-f005:**
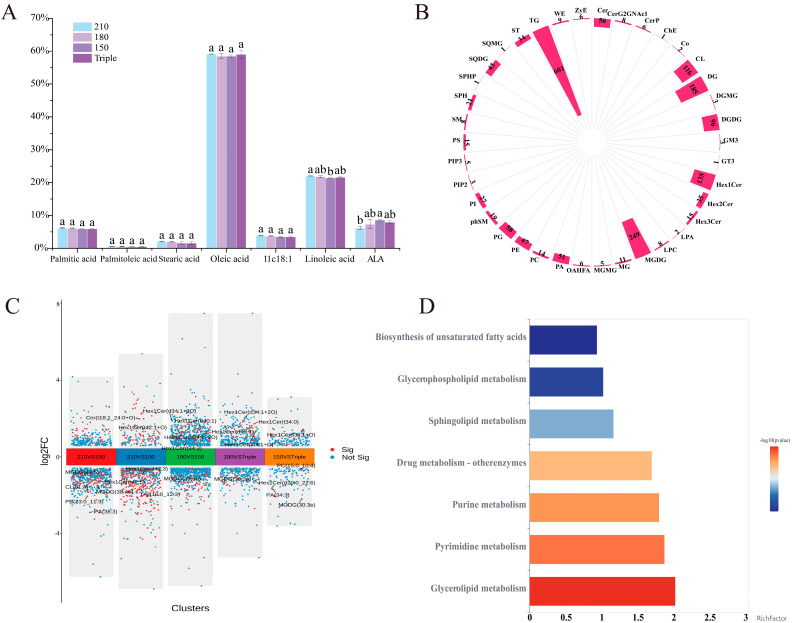
Fatty acid and lipidomics analyses of chili oil. (**A**) Column diagram of fatty acid content of four types of chili oil; (**B**) quantity diagram of chili oil quality; (**C**) differential lipid volcano diagram of chili oil; (**D**) enrichment diagram of the chili oil quality metabolic pathway. Bars with different letters represent significant difference (*p* < 0.05).

**Figure 6 foods-14-01006-f006:**
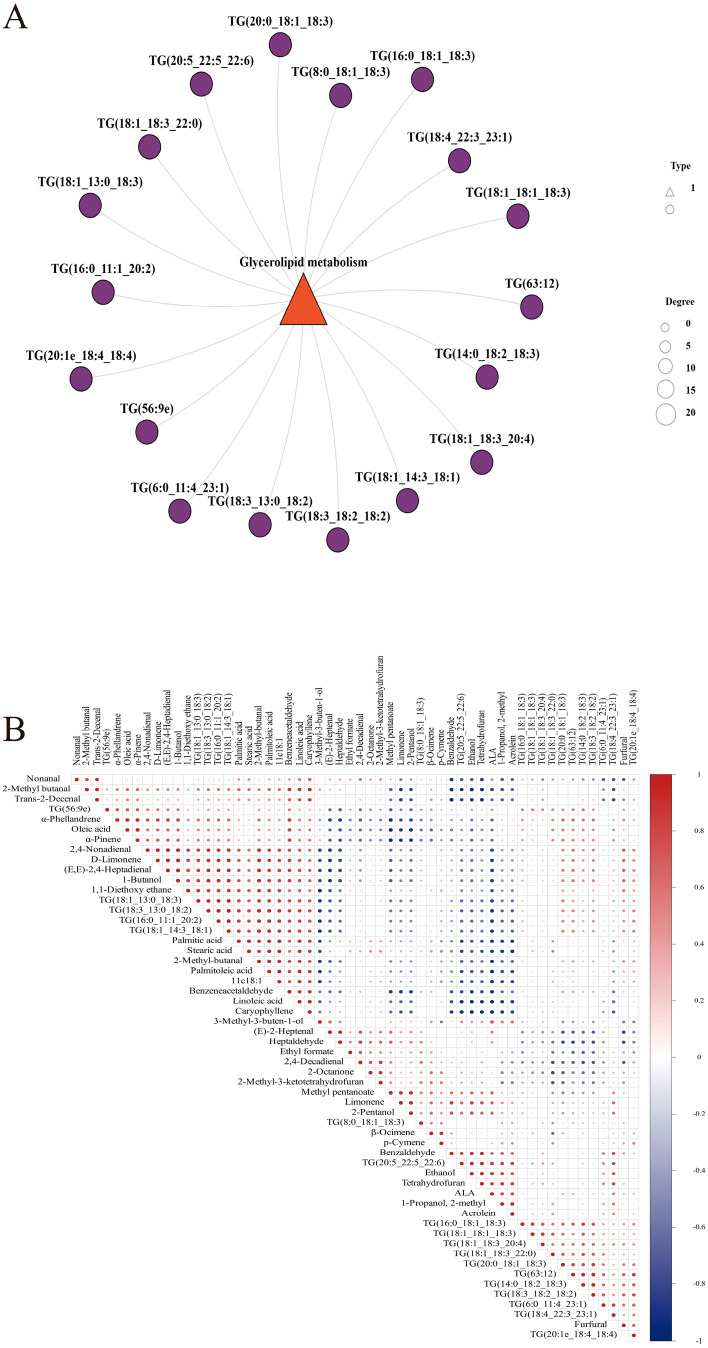
(**A**) Network diagram of key differential lipids and metabolic pathways in chili oil; (**B**) Pearson correlation analysis of 18 key differential lipids and key aroma compounds in chili oil.

**Table 1 foods-14-01006-t001:** The information of VOCs in chili oil identified by GC-IMS.

Count	Compounds	CAS	RI	Rt [Sec]	Dt [A.U.]	Signal Strength of Chili Oil Samples			
						210	180	150	Triple
1	2-Methylpyrazine	C109080	1305.5	1049.813	1.07746	4101.05 ± 367.12 ^a^	3075.07 ± 86.75 ^b^	1964.86 ± 72.03 ^d^	2460.33 ± 122.13 ^c^
2	Cyclohexanone	C108941	1304.8	1047.828	1.15969	777.05 ± 98.56 ^a^	718.52 ± 9.08 ^ab^	645.73 ± 11.46 ^b^	729.92 ± 9.03 ^a^
3	(E)-2-Heptenal-M	C18829555	1324.1	1105.601	1.25768	2319.61 ± 48.9 ^c^	2678.96 ± 27.96 ^a^	2546.42 ± 33.28 ^b^	2570.76 ± 27.33 ^b^
4	(E)-2-Heptenal-D	C18829555	1322.4	1100.43	1.66918	393.67 ± 14.83 ^d^	525.6 ± 12.54 ^a^	461.44 ± 11.33 ^c^	498.45 ± 10.83 ^b^
5	2-methyl-3-ketotetrahydrofuran-M	C3188009	1289.8	1001.287	1.07903	3161.42 ± 162.83 ^b^	3469.44 ± 39.28 ^a^	3126 ± 14.86 ^b^	3022.21 ± 80.18 ^b^
6	2-methyl-3-ketotetrahydrofuran-D	C3188009	1293.4	1013.221	1.42265	420.72 ± 9.81 ^c^	527.24 ± 31.32 ^a^	459.64 ± 11.06 ^b^	482.63 ± 23.9 ^b^
7	2-Octanone	C111137	1290.3	1002.98	1.32266	675.48 ± 34.54 ^b^	816.75 ± 68.04 ^a^	676.65 ± 28.62 ^b^	623.96 ± 41.22 ^b^
8	(E)-3-Hexen-1-ol	C928972	1304.9	1048.192	1.23114	431.53 ± 68.5 ^a^	228.16 ± 22.76 ^b^	116.58 ± 3.36 ^c^	172.71 ± 15.69 ^bc^
9	2-Hexanol	C626937	1281.1	972.99	1.28301	348.04 ± 23.95 ^a^	251.03 ± 11.37 ^b^	189.81 ± 9.03 ^c^	231.89 ± 7.49 ^b^
10	1-octanal	C124130	1280.5	971.2	1.39002	529.88 ± 24.48 ^a^	346.03 ± 13.12 ^b^	200.91 ± 10.30 ^d^	310.3 ± 14.86 ^c^
11	1-Octen-3-one	C4312996	1305	1048.361	1.27791	197.65 ± 16.12 ^a^	164.8 ± 22.17 ^b^	122.2 ± 3.79 ^c^	158.67 ± 13.87 ^b^
12	2,4-Dimethylthiazole	C541582	1267.8	929.822	1.0936	225.04 ± 9.04 ^a^	179.02 ± 8.17 ^b^	136.38 ± 7.58 ^c^	132.09 ± 10.47 ^c^
13	1-Pentanol-M	C71410	1256.6	893.263	1.25483	2146.95 ± 57.86 ^d^	3207.4 ± 28.49 ^c^	4003.41 ± 29.69 ^a^	3697.2 ± 54.22 ^b^
14	1-Pentanol-D	C71410	1256	891.335	1.51987	128.38 ± 3.33 ^d^	342.34 ± 10.71 ^c^	573.7 ± 13.00 ^a^	508.72 ± 10.79 ^b^
15	3-Methylbutyl 2-methylbutanoate	C27625350	1256.1	891.875	1.41854	178.3 ± 20.68 ^d^	284.31 ± 10.74 ^c^	374.39 ± 3.36 ^b^	395.03 ± 8.29 ^a^
16	2-Hexenal-M	C505577	1217.9	767.392	1.18068	556.03 ± 49.77 ^c^	822.58 ± 16.29 ^b^	863.82 ± 11.55 ^ab^	893.17 ± 11.87 ^a^
17	2-Hexenal-D	C505577	1217.8	767.294	1.51277	67.64 ± 3.01 ^d^	130.25 ± 4.51 ^a^	120.99 ± 1.06 ^b^	109.57 ± 2.54 ^c^
18	1-Butanol, 3-methyl-M	C123513	1207.9	734.99	1.24247	815.02 ± 113.44 ^a^	598.9 ± 24.5 ^b^	589.12 ± 14.8 ^b^	530.23 ± 24.31 ^b^
19	(+)-limonene-M	C138863	1200.7	711.651	1.2158	161.75 ± 3.09 ^b^	202.93 ± 7.12 ^a^	203.42 ± 2.95 ^a^	152.16 ± 3.46 ^c^
20	(+)-limonene-D	C138863	1200.5	710.778	1.30166	574.11 ± 15.52 ^d^	676.87 ± 9.56 ^b^	817.6 ± 7.96 ^a^	594.21 ± 11.09 ^c^
21	β-ocimene	C13877913	1193.3	687.467	1.21933	360.9 ± 12.29 ^b^	390.48 ± 11.77 ^a^	385.86 ± 8.83 ^a^	295.66 ± 1.27 ^c^
22	p-cymene	C99876	1193.1	686.726	1.30079	248.4 ± 16.86 ^a^	253.6 ± 5.95 ^a^	249.63 ± 1.58 ^a^	197.17 ± 4.85 ^b^
23	1-Butanol, 3-methyl-D	C123513	1206.9	731.873	1.48993	270.01 ± 38.97 ^a^	243.64 ± 6.20 ^ab^	246.05 ± 16.54 ^ab^	229.81 ± 6.46 ^b^
24	Heptaldehyde-M	C111717	1185.8	668.993	1.34474	1045.7 ± 15.34 ^c^	1431.67 ± 43.3 ^a^	1252.81 ± 20.03 ^b^	1254.03 ± 13.79 ^b^
25	Heptaldehyde-D	C111717	1185.8	669.016	1.69011	98.14 ± 7.73 ^c^	170.15 ± 15.97 ^b^	133.1 ± 0.75 ^b^	133.02 ± 1.30 ^a^
26	3-Methyl-3-buten-1-ol	C763326	1182.4	661.423	1.16731	243.11 ± 17.13 ^c^	299.67 ± 13.13 ^b^	317.9 ± 20.63 ^a^	353.33 ± 10.08 ^a^
27	ethyl (E)-2-butenoate	C623701	1139.8	563.984	1.18187	434.99 ± 20.05 ^d^	552.28 ± 17.38 ^b^	586.16 ± 8.86 ^a^	598.6 ± 13.65 ^c^
28	(E)-2-Pentenal-M	C1576870	1129.9	541.439	1.10678	1071.6 ± 26.25 ^d^	1972.47 ± 11.75 ^b^	2258.29 ± 7.77 ^a^	1840.85 ± 11.87 ^c^
29	(E)-2-Pentenal-D	C1576870	1129.2	539.777	1.35191	141.19 ± 5.39 ^ab^	457.17 ± 9.18 ^ab^	610.72 ± 9.14 ^b^	436.73 ± 6.72 ^a^
30	2-Pentanol	C6032297	1115.7	509.044	1.21348	187.26 ± 0.73 ^d^	220.61 ± 2.10 ^b^	237.07 ± 8.70 ^a^	193.01 ± 2.87 ^c^
31	methyl pentanoate	C624248	1104.7	483.858	1.21608	85.68 ± 6.73 ^a^	131.25 ± 3.56 ^b^	137.84 ± 2.4 ^d^	98.38 ± 1.09 ^c^
32	cyclopentanone	C120923	1100.7	474.81	1.09289	95.72 ± 2.38 ^c^	82.93 ± 1.62 ^d^	64.25 ± 2.12 ^a^	71.48 ± 0.58 ^b^
33	1-Propanol, 2-methyl-M	C78831	1094.1	462.814	1.17339	499.31 ± 21.58 ^c^	442.21 ± 7.62 ^b^	548.64 ± 5.37 ^a^	528.97 ± 7.44 ^b^
34	2-Methylpropyl propionate	C540421	1087.1	454.347	1.27644	2253.63 ± 25.34 ^a^	2847.17 ± 11.47 ^c^	2915.57 ± 12.06 ^d^	2765.34 ± 25.77 ^b^
35	1-Propanol, 2-methyl-D	C78831	1087.5	454.838	1.36941	183.51 ± 5.44 ^d^	229.49 ± 3.72 ^c^	248.04 ± 7.63 ^a^	228.19 ± 2.66 ^b^
36	Acetic acid butyl ester	C123864	1059.7	420.914	1.24241	93.78 ± 8.96 ^d^	35.99 ± 1.4 ^c^	22.55 ± 1.04 ^a^	44.75 ± 1.66 ^b^
37	Camphene	C79925	1044.6	402.469	1.19839	180.7 ± 7.37 ^d^	358.9 ± 3.33 ^c^	578.71 ± 9.07 ^a^	422.28 ± 6.84 ^b^
38	2-Methylthiophene	C554143	1044.9	402.86	1.037	784.11 ± 29.35 ^c^	1028.88 ± 0.68 ^b^	1266.67 ± 22.29 ^a^	1062.96 ± 3.50 ^a^
39	1-Propanol	C71238	1034.3	389.908	1.1132	82.79 ± 6.15 ^d^	141.97 ± 1.51 ^c^	201.64 ± 3.12 ^a^	174.81 ± 7.93 ^b^
40	2-methylpropyl 2-methylpropanoate	C97858	1044.3	402.142	1.30628	34.65 ± 2.52 ^d^	61.81 ± 6.82 ^c^	82.42 ± 5.95 ^b^	89.8 ± 6.51 ^a^
41	1-Penten-3-one-M	C1629589	1024.5	377.974	1.08163	467.39 ± 8.62 ^d^	818.54 ± 6.91 ^c^	989.66 ± 6.15 ^a^	844.12 ± 17.09 ^b^
42	1-Penten-3-one-D	C1629589	1024	377.394	1.30766	116.49 ± 6.85 ^c^	344.56 ± 4.24 ^b^	465.44 ± 7.35 ^a^	339.51 ± 2.01 ^b^
43	n-Pentanal-M	C110623	990.5	340.009	1.196	1249.3 ± 38.17 ^b^	1248.54 ± 8.91 ^b^	1317.04 ± 11.52 ^a^	1254.42 ± 8.37 ^b^
44	n-Pentanal-D	C110623	989.9	339.661	1.42194	858.62 ± 31.03 ^d^	2117.34 ± 28.65 ^b^	2272.33 ± 10.31 ^a^	1927.94 ± 28.04 ^c^
45	Ethanol-M	C64175	935.4	308.331	1.04246	932.89 ± 45.48 ^c^	1012.08 ± 29.09 ^b^	1215.26 ± 28.15 ^a^	1005.31 ± 1.74 ^b^
46	Ethanol-D	C64175	929.9	305.223	1.13009	624.97 ± 54.34 ^d^	767.03 ± 14.06 ^c^	1183.95 ± 16.76 ^a^	859.8 ± 5 ^b^
47	3-Methyl butanal	C590863	926.4	303.177	1.17748	176.05 ± 9.74 ^d^	197.59 ± 5.33 ^c^	253.62 ± 1.26 ^a^	229.49 ± 1.32 ^b^
48	1,1-diethoxy ethane	C105577	890.6	282.623	1.04629	330.27 ± 34.81 ^a^	266.04 ± 12.44 ^b^	253.46 ± 6.52 ^b^	221.97 ± 4.4 ^c^
49	Butanal-M	C123728	886.6	280.325	1.12814	446.44 ± 25.33 ^c^	502.39 ± 8.38 ^b^	550.23 ± 3.28 ^a^	533.93 ± 5.27 ^a^
50	Butanal-D	C123728	886.1	280.014	1.28234	127.97 ± 8.91 ^d^	249.64 ± 7.41 ^c^	308.59 ± 1.83 ^a^	282.97 ± 4.32 ^b^
51	Tetrahydrofuran	C109999	866.1	268.521	1.06058	96.39 ± 6.69 ^d^	109.87 ± 2.91 ^c^	387.22 ± 7.59 ^a^	189.19 ± 2.03 ^b^
52	Methyl acetate	C79209	854.2	261.688	1.02974	103.78 ± 2.6 ^d^	155.97 ± 3.00 ^c^	223.36 ± 1.38 ^a^	173.43 ± 1.36 ^b^
53	acrolein	C107028	836.7	251.653	1.06861	1098.11 ± 9.63 ^c^	1085.76 ± 18.94 ^c^	1193.93 ± 6.97 ^a^	1172.34 ± 5.6 ^b^
54	Ethyl formate	C109944	817.1	240.333	1.08588	300.26 ± 43.52 ^b^	365.94 ± 6.05 ^a^	311.61 ± 10.30 ^b^	290.63 ± 15.18 ^b^
55	1-octene	C111660	865.6	268.246	1.17378	189.96 ± 15.52 ^a^	101.87 ± 3.27 ^c^	80.84 ± 0.94 ^d^	127.26 ± 1.75 ^b^
56	Ethyl 2-methy lpropionate	C97621	950.8	317.19	1.18463	366.46 ± 15.02 ^a^	335.49 ± 6.7 ^b^	250.16 ± 1.01 ^d^	294.98 ± 3.63 ^c^
57	α-terpinolene	C586629	1280.9	972.566	1.21701	2461.53 ± 19.48 ^a^	1944.87 ± 16.09 ^b^	1410.12 ± 11.20 ^d^	1751.1 ± 9.10 ^c^
58	3-Methyl-2-butenal	C107868	1202.7	717.938	1.09149	78.43 ± 1.32 ^b^	80.77 ± 2.26 ^b^	88.24 ± 2.92 ^a^	82.28 ± 3.96 ^b^
59	Acetic acid-M	C64197	1505	1648.066	1.05986	8419.01 ± 235.96 ^a^	8657.26 ± 38.71 ^a^	8059.46 ± 44.68 ^b^	8567.82 ± 208.41 ^a^
60	Acetic acid-D	C64197	1503.7	1644.151	1.16857	6709.72 ± 617.07 ^c^	8242.27 ± 254.43 ^b^	9570.86 ± 72.06 ^a^	8011.48 ± 446.27 ^b^
61	1-Penten-3-ol	C616251	1157.2	603.728	0.9416	981.2 ± 29.22 ^d^	2018.38 ± 35.05 ^c^	2639.41 ± 19.07 ^a^	2339.27 ± 21.39 ^b^
62	2-methyl butanal-D	C96173	920.1	299.568	1.40038	7707.64 ± 60.23 ^a^	7201.29 ± 36.32 ^b^	5617.37 ± 34.45 ^c^	7155.32 ± 32.07 ^b^
63	2-methyl butanal-M	C96173	912.2	294.997	1.16278	176.26 ± 9.09 ^d^	235.59 ± 3.92 ^c^	331.1 ± 4.38 ^a^	255.76 ± 4.01 ^b^

M and D represent monomers and dimers of the same compounds, respectively. RI stands for retention index; Dt stands for drift time. Different letters in the same line indicate differences between treatment groups, *p* < 0.05. Data are mean ± SD (*n* = 3).

**Table 2 foods-14-01006-t002:** Information on VOCs in chili oil identified by GC-MS.

No.	Compounds	CAS	Content (μg/kg)			
			210	180	150	Triple
	Aldehydes					
A1	2-Methyl-butanal	96-17-3	260.50 ± 10.81 ^a^	90.50 ± 14.04 ^b^	20.25 ± 5.05 ^c^	ND
A2	2,2-Dimethyl-propanal	630-19-3	30.40 ± 4.09 ^b^	ND	ND	40.41 ± 5.05 ^a^
A3	(E)-2-Methyl-2-butenal	497-03-0	10.65 ± 2.05	ND	ND	ND
A4	(E)-2-Pentenal	1576-87-0	40.52 ± 8.09 ^a^	ND	20.52 ± 4.05 ^b^	ND
A5	Hexanal	66-25-1	10.32 ± 3.03	ND	ND	ND
A6	Furfural	98-01-1	70.55 ± 12.07 ^a^	ND	60.25 ± 8.10 ^a^	10.53 ± 1.04 ^b^
A7	(Z)-3-Hexenal	6789-80-6	90.47 ± 15.09 ^a^	10.25 ± 2.04 ^c^	10.52 ± 2.08 ^c^	50.48 ± 6.07 ^b^
A8	Heptanal	111-71-7	10.69 ± 3.12 ^b^	40.35 ± 8.05 ^a^	ND	10.58 ± 3.04 ^b^
A9	(E)-2-Heptenal	18829-55-5	20.14 ± 5.06 ^a^	10.72 ± 3.03 ^b^	ND	ND
A10	2,4-Nonadienal	6750-03-4	60.15 ± 10.05 ^a^	10.35 ± 2.04 ^b^	10.75 ± 2.02 ^b^	10.53 ± 1.02 ^b^
A11	(E,E)-2,4-Heptadienal	4313-03-5	20.85 ± 6.04	ND	ND	ND
A12	Octanal	124-13-0	20.75 ± 5.01 ^b^	30.81 ± 5.06 ^a^	ND	ND
A13	Nonanal	124-19-6	20.45 ± 2.02 ^c^	40.52 ± 6.45 ^a^	ND	30.85 ± 6.04 ^b^
A14	2-Hexenal	505-57-7	130.24 ± 15.10 ^b^	40.85 ± 7.03 ^c^	ND	490.96 ± 12.44 ^a^
A15	2-methyl-Pentanal	123-15-9	10.63 ± 3.04 ^b^	40.48 ± 6.06 ^a^	ND	10.63 ± 2.03 ^b^
A16	2-methyl-2-Butenal	107-86-8	80.75 ± 12.11 ^a^	90.78 ± 10.18 ^a^	90.85 ± 9.16 ^a^	10.85 ± 2.04 ^b^
A17	2,4-Decadienal	2363-88-4	ND	60 ± 12.09 ^a^	10.39 ± 2.04 ^b^	10.85 ± 1.03 ^b^
A18	Benzaldehyde	100-52-7	ND	ND	10.35 ± 1.03	ND
A19	Benzeneacetaldehyde	122-78-1	20.25 ± 4.04 ^a^	10.24 ± 4.04 ^b^	ND	10 ± 1.02 ^b^
A20	Trans-2-Decenal	3913-81-3	150.47 ± 10.21 ^a^	160.36 ± 10.30 ^a^	ND	160.59 ± 10.28 ^a^
Total			1050.41 ± 117.94 ^a^	630.25 ± 95.01 ^c^	230.68 ± 33.53 ^d^	840.65 ± 54.10 ^b^
	Ketones					
B1	2-Butanone	78-93-3	10.14 ± 1.04 ^d^	190.58 ± 20.23 ^b^	20.52 ± 6.05 ^c^	390.53 ± 15.42 ^a^
B2	2,2-Dimethyl-3-heptanone	19078-97-8	220.74 ± 18.31 ^a^	180.28 ± 15.25 ^b^	ND	ND
B3	2,5-Dimethyl-3-hexanone	1888-57-9	60.41 ± 8.09 ^a^	ND	10.52 ± 2.02 ^b^	ND
B4	2-Methyl-4-heptanone	626-33-5	1060.52 ± 21.22 ^a^	20.75 ± 7.04 ^c^	20.41 ± 5.06 ^c^	90.53 ± 10.14 ^b^
B5	2,4-Dimethyl-3-pentanone	565-80-0	190.89 ± 15.24 ^a^	20.24 ± 7.04 ^c^	70.52 ± 10.12 ^b^	10.85 ± 2.03 ^d^
B6	5-Methyl-3-hexanone	623-56-3	10.14 ± 1.02 ^a^	ND	ND	10.53 ± 1.05 ^a^
B7	3-Heptanone	106-35-4	50.72 ± 4.12 ^a^	ND	40.52 ± 3.08 ^b^	10.53 ± 1.04 ^c^
B8	4-Methyl-3-heptanone	6137-11-7	50.53 ± 6.14 ^a^	ND	40.52 ± 5.13 ^b^	10.56 ± 2.04 ^c^
B9	5-Nonanone	502-56-7	40.71 ± 5.14 ^b^	90.24 ± 7.18 ^a^	10.32 ± 1.04 ^c^	ND
Total			1690.15 ± 80.32 ^a^	500.28 ± 49.70 ^b^	210.21 ± 32.50 ^c^	520.23 ± 31.72 ^b^
	Esters					
C1	Propanoic acid, butyl ester	590-01-2	30.52 ± 4.04 ^b^	ND	70.56 ± 6.15 ^a^	20.85 ± 3.06 ^c^
C2	Butanoic acid, methyl ester	623-42-7	150.41 ± 10.31	ND	ND	ND
C3	2-Butenoic acid, 3-methyl-, hexyl ester	17627-41-7	10.62 ± 1.04	ND	ND	ND
C4	Propanoic acid, 2-methyl-, 2-methylpropyl ester	97-85-8	10.25 ± 1.08	ND	ND	ND
Total			200.18 ± 16.47 ^a^	ND	70.53 ± 6.15 ^b^	20.32 ± 3.06 ^c^
	Alcohols					
D1	Ethanol	64-17-5	10.41 ± 1.04 ^a^	ND	ND	10.53 ± 1.05 ^a^
D2	2-methyl-1-Butanol	137-32-6	190.72 ± 15.34 ^a^	ND	80.33 ± 8.23 ^b^	ND
D3	3-Penten-2-ol	1569-50-2	40.41 ± 3.08 ^a^	ND	10.53 ± 1.04 ^b^	ND
D4	1-Butanol	71-36-3	10.25 ± 1.06	ND	ND	ND
D5	1-Pentanol	71-41-0	10.71 ± 1.15 ^a^	ND	ND	10.11 ± 1.05 ^a^
D6	Methanethiol	74-93-1	40.52 ± 4.08 ^a^	20.34 ± 2.06 ^b^	ND	10.52 ± 1.14 ^c^
Total			300.28 ± 25.75 ^a^	20.74 ± 2.06 ^d^	90.35 ± 9.27 ^b^	30.62 ± 2.19 ^c^
	Olefins					
E1	2,4-Dimethyl- heptane	2213-23-2	180.75 ± 10.33 ^a^	70.21 ± 5.24 ^c^	ND	90.85 ± 8.23 ^b^
E2	3,4,5-Trimethyl- heptane	20278-89-1	60.73 ± 5.14 ^a^	30.26 ± 3.13 ^b^	ND	ND
E3	2,4,6-Trimethyl- heptane	2613-61-8	80.70 ± 4.19 ^b^	100.34 ± 10.24 ^a^	ND	ND
E4	1,1-Dimethoxy-2-butene	21962-24-3	80.50 ± 6.18	ND	ND	ND
E5	D-Limonene	5989-27-5	20.42 ± 3.08	ND	ND	ND
E6	2,2-Dimethyl-3-Hexene	3123-93-1	40.46 ± 4.14 ^b^	10.26 ± 1.04 ^c^	10.42 ± 1.05 ^c^	60.22 ± 5.18 ^a^
E7	1,3-Octadiene	1002-33-1	10.70 ± 1.04	ND	ND	ND
E8	Limonene	138-86-3	10.42 ± 1.03 ^b^	ND	ND	1060 ± 22.04 ^a^
E9	α-Pinene	80-56-8	30.46 ± 4.14 ^a^	10.46 ± 1.03 ^c^	10.53 ± 1.02 ^c^	20.55 ± 2.06 ^b^
E10	Camphene	79-92-5	ND	10.36 ± 1.04 ^b^	30.88 ± 1.11 ^a^	ND
E11	Caryophyllene	87-44-5	20.48 ± 3.05 ^a^	20.15 ± 2.04 ^a^	10.98 ± 1.03 ^b^	10.66 ± 0.09 ^b^
E12	α-Phellandrene	99-83-2	30.85 ± 2.07 ^a^	10.25 ± 1.02 ^c^	10.78 ± 1.14 ^c^	20.23 ± 2.07 ^b^
E13	α-terpinolene	586-62-9	ND	30.86 ± 2.12 ^a^	10.36 ± 1.09 ^b^	ND
E14	β-Ocimene	13877-91-3	20.74 ± 2.06 ^b^	20.74 ± 2.04 ^b^	40.99 ± 4.13 ^a^	10.36 ± 1.15 ^c^
Total			580.41 ± 46.45 ^b^	310.86 ± 28.94 ^c^	120.55 ± 10.57 ^d^	1270.56 ± 32.59 ^a^
	Ethers					
F1	n-Butyl ether	142-96-1	ND	10.75 ± 1.04 ^b^	20.78 ± 2.06 ^a^	ND
F2	Estragole	140-67-0	150.41 ± 12.35 ^a^	ND	20.89 ± 3.06 ^b^	ND
Total			150.42 ± 12.35 ^a^	10.75 ± 1.04 ^c^	40.56 ± 5.12 ^b^	ND
	Acids					
G1	Acetic acid	64-19-7	140.85 ± 10.34 ^a^	30.38 ± 2.19 ^b^	30.75 ± 2.10 ^b^	30.78 ± 2.08 ^b^
G2	Propanoic acid	79-09-4	390.86 ± 14.80 ^a^	220.53 ± 10.34 ^b^	100.88 ± 10.30 ^c^	10.89 ± 1.04 ^d^
G3	Pentanoic acid	109-52-4	10.96 ± 1.26	ND	ND	ND
Total			540.46 ± 26.40 ^a^	250.58 ± 12.53 ^b^	130.56 ± 12.40 ^c^	40.74 ± 3.12 ^d^
	Other compounds					
H1	Anethole	4180-23-8	10.48 ± 2.04	ND	ND	ND
H2	2-Propyl-furan	4229-91-8	450.47 ± 11.00 ^a^	10.52 ± 1.25 ^b^	10.45 ± 1.14 ^b^	450.66 ± 11.02 ^a^
H3	Trans-Bergamotene	13474-59-4	70.86 ± 6.15	ND	ND	ND
H4	p-Cymene	99-87-6	10.18 ± 1.44 ^c^	ND	20.44 ± 2.16 ^b^	30.65 ± 4.10 ^a^
Total			540.25 ± 20.63 ^a^	10.25 ± 1.25 ^d^	30.47 ± 3.30 ^c^	480.53 ± 15.12 ^b^

ND: not detected. Different letters in the same line indicate differences between treatment groups, *p* < 0.05. Data are mean ± SD (*n* = 3).

**Table 3 foods-14-01006-t003:** Volatile organic flavor compounds and ROAVs.

No.	Compounds	CAS	Threshold (In Oil) (mg/kg)	ROAV				Odor
				210	180	150	Triple	
1	2-methyl-butanal	96-17-3	0.023	100.00 ± 0.00 ^a^	100.00 ± 0.00 ^a^	84.47 ± 1.74 ^b^	-	malt
2	(E,E)-2,4-Heptadienal	4313-03-5	0.01	15.00 ± 1.25	-	-	-	fatty
3	Furfural	98-01-1	3.00	0.20 ± 0.05 ^b^	-	1.62 ± 0.04 ^a^	0.05 ± 0.02 ^c^	bread, almond, sweet popcorn
4	(E)-2-Heptenal	18829-55-5	0.013	15.38 ± 1.36 ^a^	5.77 ± 0.07 ^b^	-	-	fatty
5	2,4-Nonadienal	6750-03-4	0.005	105.00 ± 2.85 ^b^	15.00 ± 0.24 ^d^	194.29 ± 2.56 ^a^	21.51 ± 1.02 ^c^	fatty, flower, green
6	Nonanal	124-19-6	0.150	1.33 ± 0.15 ^c^	2.33 ± 0.04 ^b^	-	3.58 ± 0.25 ^a^	fatty, citrus, green
7	2,4-Decadienal	2363-88-4	0.135	-	3.89 ± 0.06 ^b^	5.40 ± 0.25 ^a^	1.19 ± 0.24 ^c^	citrus, chicken
8	Benzaldehyde	100-52-7	0.060	-	-	12.14 ± 0.15	-	nutty
9	Benzeneacetaldehyde	122-78-1	0.022	9.09 ± 0.16 ^b^	3.41 ± 0.05 ^c^	-	9.78 ± 0.26 ^a^	chocolate, cocoa
10	Trans-2-Decenal	3913-81-3	3.220	130.00 ± 1.82 ^c^	137.50 ± 1.25 ^b^	-	295.70 ± 3.25 ^a^	fatty, mushroom
11	D-Limonene	5989-27-5	0.034	4.41 ± 0.52	-	-	-	citrus, mint
12	Limonene	138-86-3	0.200	0.38 ± 0.08 ^b^	-	-	100.00 ± 0.00 ^a^	lemon, orange
13	α-Pinene	80-56-8	0.006	41.67 ± 1.89 ^c^	8.33 ± 0.25 ^d^	80.95 ± 2.56 ^a^	71.68 ± 1.58 ^b^	pine, turpentine
14	Caryophyllene	87-44-5	0.390	0.45 ± 0.07 ^c^	0.38 ± 0.02 ^d^	1.87 ± 0.05 ^a^	0.69 ± 0.02 ^b^	wood, spice
15	α-Phellandrene	99-83-2	0.040	5.63 ± 0.45 ^c^	1.25 ± 0.04 ^d^	18.21 ± 0.69 ^a^	8.06 ± 0.35 ^b^	turpentine, mint, spice
16	α-terpinolene	586-62-9	0.200	-	1.13 ± 0.09 ^b^	4.86 ± 0.21 ^a^	-	pine
17	β-Ocimene	13877-91-3	0.034	5.15 ± 0.22 ^d^	5.88 ± 0.15 ^c^	100.00 ± 0.00 ^a^	7.91 ± 0.78 ^b^	citrus, wood, green
18	p-Cymene	99-87-6	0.120	1.04 ± 0.06 ^c^	-	14.17 ± 0.74 ^a^	4.03 ± 0.35 ^b^	musty
19	1-Butanol	71-36-3	0.038	1.97 ± 0.07	-	-	-	fruit

-: not detected. The flavors and thresholds of all flavor compounds were obtained from relevant literature and websites. Data are mean ± SD (*n* = 3). Different letters in the same row indicate significant difference (*p* < 0.05).

## Data Availability

The original contributions presented in this study are included in the article. Further inquiries can be directed to the corresponding author.

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
