# Peer review of "Impact of Oil Temperature and Splashing Frequency on Chili Oil Flavor: Volatilomics and Lipidomics"

_foods, 2025, doi:10.3390/foods14061006_

Round 1

Reviewer 1 Report

Comments and Suggestions for Authors

ear Authors,

Below are some suggestions to enhance your manuscript, improving its clarity and scientific rigor to facilitate reconsideration for publication in Foods.

Abstract: please include a clearer statement of the study’s novelty and broader significance in food science

Keywords: please avoid using the same words of the title

Introduction: the introduction could better highlight the novelty of the study considering chili oil flavor

MM:

Authors didn`t utilized a control group? E.g. chili oil prepared without oil splashing to compare the effects of temperature and splashing frequency?

Please include a detailed explanation of how the 24-hour extraction period was determined. It can affect the flavor profile?

Is there an interaction between oil temperature and chili powder composition?

Results and Discussion

Authors think that the findings could be generalizable to other chili varieties or oil types?

There is a potential impact of chili powder particle size and storage conditions on flavor development?

Other lipid classes (e.g., phospholipids, sterols) could contribute to flavor formation?

Please, if present, discuss the limitations of the study

Please, better integrate the findings from the lipidomics analysis with the VOCs to provide a more cohesive discussion. 

The study could better emphasize the practical applications of its findings.

The discussion is a slight overstatement of the role of glycerolipid metabolism in flavor formation. Authors could include other metabolic pathways that also contribute to flavor

The content of certain compounds in Table 1 does not always align with the descriptions in the text

Figure 1 could include a brief explanation of the zones (I, II, III) in the caption.

Are there potential health implications of the identified volatile compounds and lipid oxidation?

The study does not discuss consumer perception or sensory analysis 

Some comparisons lack statistical significance testing (p-values).

Conclusion: broader implications of the findings for the food industry are missing (industry professionals or consumers could apply this knowledge?)

Author Response

Below are some suggestions to enhance your manuscript, improving its clarity and scientific rigor to facilitate reconsideration for publication in Foods.

Abstract: please include a clearer statement of the study’s novelty and broader significance in food science

Response: Thanks for your comments, We have rewritten the abstract to increase the innovation of the paper and its application value in the food industry (line 26-29).

Keywords: please avoid using the same words of the title.

Response: We have changed the keywords (line 30).

Introduction: the introduction could better highlight the novelty of the study considering chili oil flavor

Response: We have made changes to the introduction according to your suggestion (line 85-88).

MM:

Authors didn`t utilized a control group? E.g. chili oil prepared without oil splashing to compare the effects of temperature and splashing frequency?

Response: We appreciate this comment. The control group we had in this study was the triple splashing group, which was made by the traditional red oil making procedure.

We did test the chili oil prepared without oil splashing by soaking chili into oil at room temperature for 24 hrs in our preliminary study, it gave very similar flavor profile to the oil itself, with a hint of chili smell, which was far away from our interested signature red oil flavor. The prelim results suggested heated oil was critical for our red oil flavor formation. Since the objective of this study was to reveal the chemistry of signature red oil flavor formation, we intended to exclude the chili oil prepared without oil splashing group, to better focus on the differences in the effects of different oil temperatures and three oil splashes versus a single splash on the flavor of the chili oil.

Please include a detailed explanation of how the 24-hour extraction period was determined. It can affect the flavor profile?

Response: We appreciate this constructive comment to refine our manuscript. The 24 hrs was determined primarily by traditional protocols, and it was also shown in previous studies to be a good time to balance flavor generation and lipid oxidation. More detailed was explained in Line 66-80. Thermal oil extraction induces lipid oxidation, characterized by hydroperoxide formation and degradation into flavor-active compounds such as aldehydes, ketones, alcohols, acids, and furans [1]. Research indicates that immersing chili oil in beef tallow for 24 hours can enhance the capsaicin content by 15.8% [2], while immersing five-spice flavored oil for over 18 hours significantly promotes the dissolution of volatile flavor compounds [3]. Although prolonging the immersion time can enhance the flavor quality, extending it beyond 24 hours results in an increased peroxide value [4]. Given that under normal temperature, lipids remain in the initial stage of oxidation (with a lower peroxide value) within 24 hours [5], precisely controlling the immersion time of chili oil to 24 hours can both maximize the dissolution of flavor compounds and effectively inhibit excessive oxidation, thereby optimizing the overall quality of the product.

[1] Xu, B.; Zhang, L.; Ma, F.; Zhang, W.; Wang, X.; Zhang, Q.; Luo, D.; Ma, H.; Li, P. Determination of free steroidal compounds in vegetable oils by comprehensive two-dimensional gas chromatography coupled to time-of-flight mass spectrometry. Food Chemistry 2018, 245, 415-425, doi:https://doi.org/10.1016/j.foodchem.2017.10.114.

[2]Li, D.; He, X.; Li, Y.; Yao, D. Study on processing technology of chili flavor beef tallow andits volatile compounds analysis. Food&Machinery 2021, 37, 149-154,214, doi:10.13652/j.issn.1003-5788.2021.12.025.

[3]Xu, L.; Sun, P.; Yu, X.; Qu, Q.; Zhang, Z. Optimization of processing conditions for five-spice condiment oil based on electronic nose analysis. Food Science 2014, 35, 308-313, doi:10.7506/spkx1002-6630-201420060.

[4]Mishra, S.; Firdaus, M.A.; Patel, M.; Pandey, G. A study on the effect of repeated heating on the physicochemical and antioxidant properties of cooking oils used by fried food vendors of Lucknow city. Discover Food 2023, 3, 7, doi:10.1007/s44187-023-00046-8.

[5]Liu, M.; Hu, L.; Deng, N.; Cai, Y.; Li, H.; Zhang, B.; Wang, J. Effects of different hot-air drying methods on the dynamic changes in color, nutrient and aroma quality of three chili pepper (Capsicum annuum L.) varieties. Food Chemistry: X 2024, 22, 101262, doi:https://doi.org/10.1016/j.fochx.2024.101262.

Is there an interaction between oil temperature and chili powder composition?

Response: Yes, there could be an interaction between oil temperature and chili powder composition. But the interaction was could be ignored at room temperature, as in our preliminary test the chili soaking oil had no signature red oil flavor. The interaction could be very important when it comes to heated conditions. More detailed was explained in Line 89-100. 

Chili powder contains proteins, fats, and reducing sugars [1], which are prone to lipid oxidation and Maillard reactions under heating conditions, forming various volatile organic compounds [2]. During this process, proteins undergo thermal degradation and participate in Maillard reactions, generating flavor compounds such as pyrazines and pyrroles. The increase of nitrogen-containing compounds (such as pyrroles) found in the study by Ye et al. [3] could possibly be originate from the thermal degradation of proteins. Meanwhile, heating also leads to extensive degradation of fatty acids [4]. During the frying process, unsaturated fatty acids are broken down into aldehydes, ketones, furans, and pyridine compounds [1], and interact with Maillard reaction products [5]. In addition, carbohydrates (such as reducing sugars) undergo caramelization at high temperatures (140-180°C), producing furans, aldehydes (such as isovaleraldehyde and 2-methylbutanal), and other key aroma compounds in chili oil [6].

[1]Liu, M.; Hu, L.; Deng, N.; Cai, Y.; Li, H.; Zhang, B.; Wang, J. Effects of different hot-air drying methods on the dynamic changes in color, nutrient and aroma quality of three chili pepper (Capsicum annuum L.) varieties. Food Chemistry: X 2024, 22, 101262, doi:https://doi.org/10.1016/j.fochx.2024.101262.

[2]Diez-Simon, C.; Ammerlaan, B.; van den Berg, M.; van Duynhoven, J.; Jacobs, D.; Mumm, R.; Hall, R.D. Comparison of volatile trapping techniques for the comprehensive analysis of food flavourings by Gas Chromatography-Mass Spectrometry. Journal of Chromatography A 2020, 1624, 461191, doi:https://doi.org/10.1016/j.chroma.2020.461191.

[3]Ye, M.; Wang, J.; Xu, H.; Li, M. Effect of oil temperature on the quality of chili oil. China Condiment 2022, 47, 124-127,132, doi:10.3969/j.issn.1000-9973.2022.01.024.

[4]Huang, W.; Liu, Q.; Fu, X.; Wu, Y.; Qi, Z.; Lu, G.; Ning, J. Fatty acid degradation driven by heat during ripening contributes to the formation of the “Keemun aroma”. Food Chemistry 2024, 451, 139458, doi:https://doi.org/10.1016/j.foodchem.2024.139458.

[5]Guo, D.; Wan, P.; Liu, J.; Chen, D.-W. Use of egg yolk phospholipids to boost the generation of the key odorants as well as maintain a lower level of acrylamide for vacuum fried French fries. Food Control 2021, 121, 107592, doi:https://doi.org/10.1016/j.foodcont.2020.107592.

[6]Lin, S.; Ma, W.; He, X.; Fu, G.; Zhong, J.; Peng, H.; Wan, Y. Effect of oil temperatures on the flavor and spiciness of chili oil. Journal of Henan University of Technology(Natural Science Edition) 2023, 44, 25-32, doi:10.16433/j.1673-2383.2023.05.004.

Results and Discussion

Authors think that the findings could be generalizable to other chili varieties or oil types?

Response: Thank you for the very inspiring comment, we have added the answer into our Conclusion (Line 712-717).In short, our findings could be generalizable to other chili varieties, but not other oil types. Because red oil flavor was mainly generated from Maillard reaction and lipid transformation. Different chili varieties have very similar sugars, proteins, fats profiles, so the flavors generated via Millard reaction would not alter much among different varieties. But different oil types usually have distinguished lipid compositions, which can result in a huge variation in the flavors generated via lipid transformation.

More detailed explanation are as follows. Wang [1] investigated the contents of carotenoids, capsaicinoids, sugars, fats, and proteins in 9 diverse varieties of dried chili peppers, including Neihuang New Generation and Chaotianjiao. Significant variations were detected in the total carotenoid and capsaicinoid contents among the different chili pepper varieties. Notably, the Neihuang New Generation chili pepper exhibited a carotenoid content exceeding 300 mg/100 g and a capsaicinoid content exceeding 20 mg/100 g, which were substantially higher than those in other varieties. However, the primary fatty acid composition, total sugar content, and protein content remained largely consistent across the different chili pepper varieties. Yang [2] performed a flavor analysis of chili oils derived from various dried chili peppers and observed a similarity in flavor profiles between the Bullet Head chili pepper and Longji chili pepper. Consequently, the primary differences among the chili pepper varieties are primarily in the contents of capsaicinoids and carotenoids, while the differences in protein, fat, and carbohydrate contents are negligible, resulting in similar Maillard reaction outcomes and flavor compositions. Hence, our research on the influence of oil temperature on the flavor of chili oil can be generalized to different chili pepper varieties. Nevertheless, owing to the substantial differences in fatty acid composition among various oils [3], Yang [4] reported significant disparities in the capsaicinoid content, color difference, peroxide value, and volatile organic compound contents of chili oils produced from rapeseed oil, corn oil, and soybean oil. Similarly, Tao [5] also noted significant differences in the volatile component compositions of chili oils prepared with different types of edible oils. Given that the flavor compounds generated by different oils may vary considerably, our study may not be generalizable to other types of oils. 

[1]Wang, X.; Rao, L.; Wang, Y.; Wu, X.; Zhao, L.; Liao, X. Quality analysis and evaluation of nine vanieties of dried peppers. Science and Technology of Food Industry 2022, 43, 300-310, doi:10.13386/j.issn1002-0306.2022010251.

[2]Yang, F.; Yuan, H.; Jia, H.; Deng, F.; Wang, Z. Effects of chili varieties on physicochemical properties and flavor compounds of chili oil based on GC-IMS combined with multivariate statistical methods. Food and Fermentation Industries 2023, 49, 319-328, doi:10.13995/j.cnki.11-1802/ts.033362.

[3]Machado, M.; Rodriguez-Alcalá, L.M.; Gomes, A.M.; Pintado, M. Vegetable oils oxidation: mechanisms, consequences and protective strategies. Food Reviews International 2023, 39, 4180-4197, doi:10.1080/87559129.2022.2026378.

[4]Yang, F.; Wang, X.; Jia, H.; Xu, C.; Yuan, H. Comparison of the flavor qualities of chili oils prepared from different types of vegetableoil. Modern Food Science and Technology 2024, 40, 338-350, doi:10.13982/j.mfst.1673-9078.2024.10.0985.
[5]Tao, X.; Wang, Y.; Yin, S.; Liu, C.; Liu, J.; Zhang, Q.; Liu, D.; Zhou, P. Effects of different types of cooking oil on flavor characteristics of chili oil. Food and Fermentation Industries 2024, 50, 341-352, doi:10.13995/j.cnki.11-1802/ts.040210.

There is a potential impact of chili powder particle size and storage conditions on flavor development?

Response: Thanks for pointing this out. Yes, the chili powder particle size and storage condition could have impact on flavor development. That’s why we grounded the chili in to the same size and let the oil stored in the identical condition in our study (Line 140-141, 149-151).

The particle size of chili powder is a critical factor influencing the quality and flavor of chili oil. Yang et al. [1] investigated the impact of various chili powder particle sizes on the physicochemical properties and volatile flavor compounds in chili oil, discovering that a reduction in particle size enhances the dissolution of capsaicin and dihydrocapsaicin. Similar findings were reported by Chen et al. [2], who also identified that a particle size of 60 mesh optimizes the quality of chili oil.

Storage conditions such as temperature, light exposure, oxygen, and storage duration significantly affect the flavor of chili oil. High temperatures and light, particularly ultraviolet rays, accelerate oxidation and degrade flavor compounds [3]. Terpenoids, abundant in chili oil, are highly sensitive to oxygen and ultraviolet radiation, leading to oxidation and the production of volatile components like alcohols, aldehydes, and ketones [4]. For instance, orange oil exposed to sunlight (ultraviolet radiation) develops a strong oily off-flavor [5], and the levels of limonene and linalool in pepper oil significantly decrease. Thus, sealing, light avoidance, and low-temperature storage are essential to mitigate flavor degradation in chili oil.

Research indicates that the concentration of limonene and linalool in pepper oil declines with extended storage time [3], suggesting that proper storage duration is vital for preserving flavor compounds.

[1]Yang, F.; Deng, F.; Jia, H.; Yuan, H.; Yao, K. Study on the effects of granularity of paprika on physicochemicalproperties and volatile flavor compounds of chili oil. Food&Machinery 2023, 39, 157-165, doi:10.13652/j.spjx.1003.5788.2023.80066.

[2]Chen, L.; Yuan, C.; Jiang, H.; Wu, H.; Qiao, M.; Yang, F. Study on effect of chili particle size on the quality and flavor of chili oil. China Condiment 2023, 48, 63-66,74, doi:10.3969/j.issn.1000-9973.2023.03.011.

[3]Sun, J.; Sun, B.; Ren, F.; Chen, H.; Zhang, N.; Zhang, Y.; Zhang, H. Effects of Storage Conditions on the Flavor Stability of Fried Pepper (Zanthoxylum bungeanum) Oil. Foods 2021, 10, 1292, doi:10.3390/foods10061292.

[4]Wibowo, S.; Grauwet, T.; Kebede, B.T.; Hendrickx, M.; Van Loey, A. Study of chemical changes in pasteurised orange juice during shelf-life: A fingerprinting-kinetics evaluation of the volatile fraction. Food Research International 2015, 75, 295-304, doi:https://doi.org/10.1016/j.foodres.2015.06.020.

[5]&nbspSun, H.; Ni, H.; Yang, Y.; Wu, L.; Cai, H.N.; Xiao, A.F.; Chen, F. Investigation of sunlight-induced deterioration of aroma of pummelo (Citrus maxima) essential oil. J Agric Food Chem 2014, 62, 11818-11830, doi:10.1021/jf504294g.

Other lipid classes (e.g., phospholipids, sterols) could contribute to flavor formation?

Response: Many thanks. Yes, other lipid classes (e.g., phospholipids, sterols) could contribute to flavor formation, according to previous studies. Phospholipid oxidation degradation yields key flavor compounds. For instance, egg yolk phospholipids can enhance the formation of flavor compounds like (E,E)-2,4-decadienal during frying [1]. β-Sitosterol, campesterol, and stigmasterol are the predominant phytosterols in red peppers [2], and vegetable oils also contain a rich concentration of sterol compounds [3]. Presently, the research on the impact of sterol compounds on flavor is scarce, indicating a need for additional studies.

However, these other lipid classes (e.g., phospholipids, sterols) were not recognized as major flavor contributors by our statistical methods (Figure 5C). To keep our contents focused on the flavor contributors, the authors would prefer to not to distract readers by involving too much information.

[1]Guo, D.; Wan, P.; Liu, J.; Chen, D.-W. Use of egg yolk phospholipids to boost the generation of the key odorants as well as maintain a lower level of acrylamide for vacuum fried French fries. Food Control 2021, 121, 107592, doi:https://doi.org/10.1016/j.foodcont.2020.107592.

[2]Choi, E.; Chun, H.S.; Auh, J.-H.; Ahn, S.; Kim, B.H. Evaluation of sterols as markers of fungal spoilage in red pepper powder. Food Chemistry 2024, 452, 139566, doi:https://doi.org/10.1016/j.foodchem.2024.139566.

[3]Schlag, S.; Schäfer, S.; Sommer, K.; Vetter, W. A sterol database: GC/MS data and occurrence of 150 sterols in seventy-four oils. Food Chemistry 2024, 460, 140778, doi:https://doi.org/10.1016/j.foodchem.2024.140778.

Please, if present, discuss the limitations of the study

Response: Thank you very much. We have added the limitation of the study in Line 712-717.

Although our findings could be generalizable to other chili varieties with similar sugars, proteins, fats profiles, the whole picture of chili oil flavor development would be more comprehensively presented if the effect of other oil types were investigated. Future works may dig on revealing the molecular mechanisms of specific lipid degradation pathways (e.g., oxidation, enzymatic degradation) and verification of the generation route of the key aroma compounds.

Please, better integrate the findings from the lipidomics analysis with the VOCs to provide a more cohesive discussion. 

Response: As suggested, we complemented a more in-depth discussion of the correlation between lipidomics and key flavor substances in section 3.6, Line 645-662, 681-685.

The study could better emphasize the practical applications of its findings.

Response: Thank you! We have added the practical applications of this study in the conclusion, line 707-711.

The study clarified the role of different oil temperatures (210°C, 180°C, 150°C) on the regulation of key flavor substances (e.g., aldehydes, ketones, and terpenes) in chili oil, and provided temperature control standards for industrial production. By adjusting the splashing temperature or adopting a staged temperature treatment (e.g., three-stage splashing), consumers and industry professionals may customize chili oil products with different aroma profiles.

The discussion is a slight overstatement of the role of glycerolipid metabolism in flavor formation. Authors could include other metabolic pathways that also contribute to flavor.

Response: We sincerely appreciate this valuable comment! As suggested, we included other glycerophospholipid and sphingolipid pathways into discussion (Line 621-631). Pan [1] showed that glycerophospholipid metabolism is the main pathway of lipid oxidation in hazelnut oil. The unsaturated fatty acids generated from glycerophospholipid metabolism was reported to serve as crucial precursors for the synthesis of aromatic compounds [2]. Li [3] discovered that sphingolipid metabolism was the most significantly enriched lipid metabolic pathway during the storage oxidation of sea buckthorn fruit oil. Additionally, the decrease in sphingolipid metabolites was observed in oil during high-temperature or frying processes [3]. Yu [4] explored the potential link between the lipids and flavors of watermelon seeds and identified sphingolipid metabolism as the major metabolic pathway, as indicated by KEGG metabolic pathway enrichment analysis.

[1]Pan, L.; Xu, W.; Gao, Y.; Ouyang, H.; Liu, X.; Wang, P.; Yu, X.; Xie, T.; Li, S. Exploring the lipid oxidation mechanisms during pumpkin seed kernels storage based on lipidomics: From phenomena, substances, and metabolic mechanisms. Food Chemistry 2024, 455, 139808, doi:https://doi.org/10.1016/j.foodchem.2024.139808.

[2]Gao, C.; Li, Q.; Wen, H.; Zhou, Y. Lipidomics analysis reveals the effects of Schizochytrium sp. supplementation on the lipid composition of Tan sheep meat. Food Chemistry 2025, 463, 141089, doi:https://doi.org/10.1016/j.foodchem.2024.141089.

[3]Li, Y.; Wan, Y.; Wang, J.; Zhang, X.; Leng, Y.; Wang, T.; Liu, W.; Wei, C. Investigation of the oxidation rules and oxidative stability of seabuckthorn fruit oil during storage based on lipidomics and metabolomics. Food Chemistry 2025, 476, 143238, doi:https://doi.org/10.1016/j.foodchem.2025.143238.

[4]Yu, X.; Li, B.; Ouyang, H.; Xu, W.; Zhang, R.; Fu, X.; Gao, S.; Li, S. Exploring the oxidative rancidity mechanism and changes in volatile flavors of watermelon seed kernels based on lipidomics. Food Chemistry: X 2024, 21, 101108, doi:https://doi.org/10.1016/j.fochx.2023.101108.

The content of certain compounds in Table 1 does not always align with the descriptions in the text

Response: Sorry for the confusion. We have checked the corresponding data in Table 1 and corrected them at the corresponding place in the text.

Figure 1 could include a brief explanation of the zones (I, II, III) in the caption.

Response: We have added description of the regions in the legend of Figure 1.

Are there potential health implications of the identified volatile compounds and lipid oxidation?

Response: Yes, there are some potential beneficial compounds. But to keep the contents focus on oil flavor and its development chemistry, we did not discuss the potential health implications. This could be a very nice direction for the future study.

Some beneficial volatiles and lipids are listed as follow. Acrolein, a byproduct of glycerol thermal degradation, when given to rats at a dose of 5 mg/kg over a long period, can decrease the release of excitatory neurotransmitters and harm the rats’ memory and learning abilities. Fried foods contribute to higher acrolein levels in the diet [1]. Additionally, 2-hexenal, a lipid oxidation product, has mutagenic, genotoxic, and carcinogenic effects and may contribute to cardiovascular diseases [2]. Despite the potential health risks of acrolein and 2-hexenal, the levels in chili oil and the quantity consumed suggest that the health risks are manageable. Research also indicates that reheating rapeseed oil frequently can generate significant amounts of trans fats and highly reactive oxidative products, potentially increasing the risk of cardiovascular diseases [3]. However, the process of making chili oil involves brief frying (less than 5 minutes), and studies show that due to its high content of polyunsaturated fatty acids, rapeseed oil remains stable during such short frying periods [4]. This stability reduces the potential health risks from the primary and secondary oxidation products formed during chili oil production. 

[1]Jiang, K.; Huang, C.; Liu, F.; Zheng, J.; Ou, J.; Zhao, D.; Ou, S. Origin and Fate of Acrolein in Foods. Foods 2022, 11, doi:10.3390/foods11131976.

[2]Grootveld, M.; Percival, B.C.; Leenders, J.; Wilson, P.B. Potential Adverse Public Health Effects Afforded by the Ingestion of Dietary Lipid Oxidation Product Toxins: Significance of Fried Food Sources. Nutrients 2020, 12, doi:10.3390/nu12040974.

[3]Multari, S.; Marsol-Vall, A.; Heponiemi, P.; Suomela, J.-P.; Yang, B. Changes in the volatile profile, fatty acid composition and other markers of lipid oxidation of six different vegetable oils during short-term deep-frying. Food Research International 2019, 122, 318-329, doi:https://doi.org/10.1016/j.foodres.2019.04.026.

[4]Liu, W.; Luo, X.; Huang, Y.; Zhao, M.; Liu, T.; Wang, J.; Feng, F. Influence of cooking techniques on food quality, digestibility, and health risks regarding lipid oxidation. Food Research International 2023, 167, 112685, doi:https://doi.org/10.1016/j.foodres.2023.112685.

The study does not discuss consumer perception or sensory analysis 

Response: Thanks for the thoughtful comment. The objective of this study was to reveal the changes of flavor-related compounds in the oil preparation. Consumer perception or sensory analysis would be of great interest to us, especially for the product development, but not the priority of this study. As compensation of sensory test, this study utilized electronic noses and electronic tongues, to assess the flavor characteristics of chili oil and provide some overall sensory information. In the future, the authors plan to conduct sensory evaluations with panelists to examine the influence of different oil splashing durations on the flavor profile of chili oil.

Some comparisons lack statistical significance testing (p-values).

Response:  Tons of thanks, we have added and labeled the significance test in Table 2 and Table 3.

Conclusion: broader implications of the findings for the food industry are missing (industry professionals or consumers could apply this knowledge?)

Response: The application prospective comment is highly appreciated!As suggested, we added the broader implications of the findings for the food industry in line 707-711.

The study clarified the role of different oil temperatures (210°C, 180°C, 150°C) on the regulation of key flavor substances (e.g., aldehydes, ketones, and terpenes) in chili oil, and provided temperature control standards for industrial production. By adjusting the splashing temperature or adopting a staged temperature treatment (e.g., three-stage splashing), consumers and industry professionals may customize chili oil products with different aroma profiles.

Reviewer 2 Report

Comments and Suggestions for Authors

The manuscript by Li et al. presents a meticulously executed and insightful investigation into the formation of key aroma compounds in chilli oil, analyzing the effects of varying oil temperatures and splashing processes. The authors utilize a combination of advanced techniques, including headspace-gas chromatography-ion mobility spectrometry, headspace gas chromatography-mass spectrometry, and lipidomics, to comprehensively analyze the volatile and lipid profiles of chilli oil. Their research identifies 31 key aroma compounds and reveals significant correlations between specific fatty acids and the formation of these critical aromatic components. Such findings offer valuable insights into how oil temperature and splashing frequency influence the aroma profile of chilli oil. Integrating advanced spectrometric methods provides a robust framework to enhance our understanding of aroma formation in chilli oil. The scientific significance of this work is considerable, contributing to the broader fields of food chemistry and flavour science. Moreover, the study’s results are promising for optimizing chilli oil production to achieve superior aroma profiles. The statistical analyses presented in the manuscript are well-structured and appropriately executed. 
However, there are a few areas where minor improvements could enhance the overall clarity and impact of the study: 
1. For the untargeted metabolomics analysis, it would be beneficial to adhere to the minimum reporting standards for LC-MS as outlined in the following manuscript: Alseekh, S. et al. (2021). “Mass spectrometry-based metabolomics: a guide for annotation, quantification and best reporting practices.” Nat. Met. 18, 747-756. 
2. Clarifying whether the identified key aroma compounds are exclusive to chilli oil or if they are also present in other cooking oils would add depth to the analysis. 
3. Providing additional context on how these notable findings can be applied within industrial food processing would significantly enhance the practical relevance of the study. These suggestions aim to refine a manuscript further that, despite its considerable strengths, could benefit from increased clarity and contextual depth.

Author Response

The manuscript by Li et al. presents a meticulously executed and insightful investigation into the formation of key aroma compounds in chilli oil, analyzing the effects of varying oil temperatures and splashing processes. The authors utilize a combination of advanced techniques, including headspace-gas chromatography-ion mobility spectrometry, headspace gas chromatography-mass spectrometry, and lipidomics, to comprehensively analyze the volatile and lipid profiles of chilli oil. Their research identifies 31 key aroma compounds and reveals significant correlations between specific fatty acids and the formation of these critical aromatic components. Such findings offer valuable insights into how oil temperature and splashing frequency influence the aroma profile of chilli oil. Integrating advanced spectrometric methods provides a robust framework to enhance our understanding of aroma formation in chilli oil. The scientific significance of this work is considerable, contributing to the broader fields of food chemistry and flavour science. Moreover, the study’s results are promising for optimizing chilli oil production to achieve superior aroma profiles. The statistical analyses presented in the manuscript are well-structured and appropriately executed. 

Thank you very much for the positive comments!

However, there are a few areas where minor improvements could enhance the overall clarity and impact of the study: 

  1. For the untargeted metabolomics analysis, it would be beneficial to adhere to the minimum reporting standards for LC-MS as outlined in the following manuscript: Alseekh, S. et al. (2021). “Mass spectrometry-based metabolomics: a guide for annotation, quantification and best reporting practices.” Nat. Met. 18, 747-756. 

Response: Thanks for your suggestion. After carefully checking of the paper and our data processing procedure, We  cited the referred literature into the methods section (line 262-263).

  1. Clarifying whether the identified key aroma compounds are exclusive to chilli oil or if they are also present in other cooking oils would add depth to the analysis. 

Response: Thank you! As suggested, the key aroma compounds are exclusive to chili oil was discussed in line 511-516.

Previous study found that the concentration of primary terpenes, including α-pinene and α-terpinolene, diminished during the heating of rapeseed oil [1]. Yet in rapeseed oil-based chili oil in this study, primary terpenes such as α-phellandrene, β-ocimene, and limonene were identified as essential flavor components, indicating these terpenes were not generated by heating rapeseed oil but were uniquely from chili oil. Aldehydes were another class of critical volatiles.

[1]Multari, S.; Marsol-Vall, A.; Heponiemi, P.; Suomela, J.-P.; Yang, B. Changes in the volatile profile, fatty acid composition and other markers of lipid oxidation of six different vegetable oils during short-term deep-frying. Food Research International 2019, 122, 318-329, doi:https://doi.org/10.1016/j.foodres.2019.04.026.

  1. Providing additional context on how these notable findings can be applied within industrial food processing would significantly enhance the practical relevance of the study. These suggestions aim to refine a manuscript further that, despite its considerable strengths, could benefit from increased clarity and contextual depth.

Response: We sincerely appreciate this constructive comment! We added the practical relevance of our findings for the food industry in Abstract (line 29) and Conclusion (line 707-711).

The present study reveals how sequential oil splashing processes synergistically activate distinct lipid degradation pathways compared to single-temperature treatments, providing new insights on lipid-rich condiments preparation, enabling chefs and food manufacturers to target specific aroma profiles.(line 29)

The study clarified the role of different oil temperatures (210°C, 180°C, 150°C) on the regulation of key flavor substances in chili oil, and provided temperature control standards for industrial production. By adjusting the splashing temperature or adopting a staged temperature treatment, consumers and industry professionals may customize chili oil products with different aroma profiles

Round 2

Reviewer 1 Report

Comments and Suggestions for Authors

The authors responded to my suggestions and they have addressed all the comments appropriately. In my opinion, the manuscript is now ready for acceptance.